# MAE-Pure: Semantic-Preserving Adversarial Purification

## Abstract

Adversarial purification is a category of defense techniques that use a generative model to eliminate adversarial perturbations. In pursuit of high performance in the cleansing of adversarial examples, current methods prefer powerful generative models (typically a diffusion model). This study investigates the purification from a novel perspective of preserving semantic relationships among image patches. Our method leverages Mask Autoencoder (MAE), which yields superior performance. Specifically, from both theoretical and experimental analysis, we disclose that the reconstruction performance of MAE is highly susceptible to adversarial noise, since the semantic relationships among patches will change significantly. Based on this intriguing property, we propose a purification scheme, named MAE-Pure, which purifies noises by preserving patch semantic relationships. We prove that this mechanism can be transformed into one tractable optimization problem with respect to the input image. Furthermore, we build a robust MAE-Pure by finetuning the purification model by introducing classification loss to further certify the patch semantic relationships. Additionally, we adapt our insight on mask diffusion model which embodies powerful generative capability to reinforce our method. A series of experiments demonstrate the superiority of our method, achieving new state-of-the-art results.

## 1 Introduction

Deep Neural Networks (DNNs) are vulnerable to adversarial examples [5, 35, 12, 25], which are imperceptible to humans. However, these inputs with the malicious perturbations can cause DNNs to make erroneous predictions. Adversarial training [26, 47] is the state-of-the-art (SOTA) method for defending against adversarial attacks. However, the tradeoff between generalization and robustness remains a concern [47], especially against unseen adversarial examples. Furthermore, adversarial training incurs significantly higher computational costs compared to standard training.

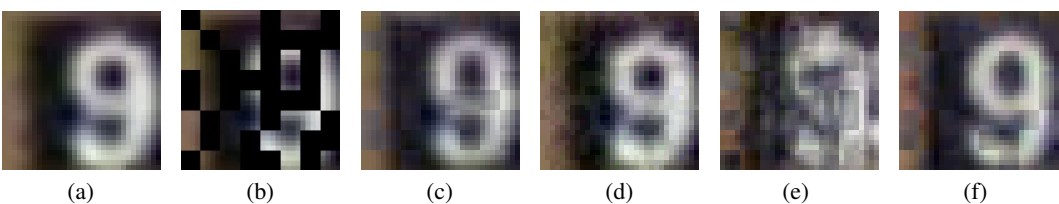

Figure 1: (a) Original image, (b) Masked image, (c) Clean image reconstruction from MAE, (d) Adversarial example under AutoAttack, (e) Reconstruction of adversarial example under AutoAttack from MAE, (f) Reconstruction of the denoised image under AutoAttack from MAE (denoised by our MAE-Pure).

Submitted to 39th Conference on Neural Information Processing Systems (NeurIPS 2025). Do not distribute.

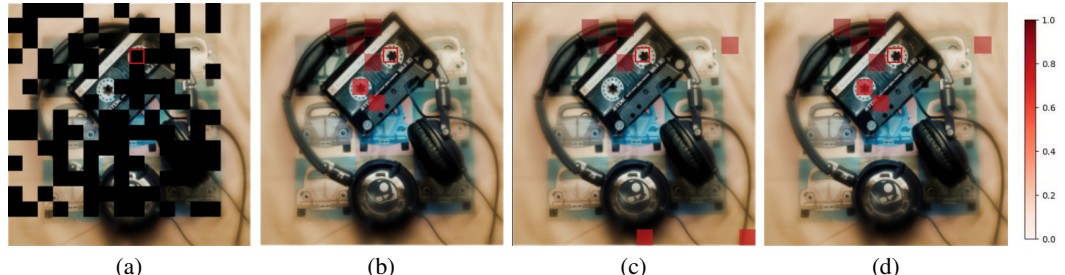

| (a) | (b) | (c) | (d) |

Figure 2: The first column, Fig. (a) represents the Mask Matrix. The second column, Fig. (b) illustrates the Attention Weights for clean samples. The third column, Fig. (c)depicts the Attention Weights for adversarial examples. The fourth column, Fig. (d) showcases the Attention Weights for denoised samples (by our MAE-Pure). Patches with a deeper red color mean the elements with more attention. The data is sampled from the ImageNet dataset [11].

Alternatively, another notable defense strategy is adversarial purification, which attracts widespread attention. Adversarial purification can be broadly classified into two categories, including purification with generative models [43, 29, 22, 3, 48] and adaptation-based purification [34]. Generative model-based approaches are the most widely used methods in adversarial purification, typically harnessing the powerful capabilities of generative models (e.g., diffusion) to transform the distribution of adversarial examples to that of clean samples [29]. Future efforts will aim to further enhance the denoising capabilities of the purification model through various approaches. These include leveraging contrastive guidance to steer diffusion models [3], integrating classifier confidence guidance into the denoising process [48], and fine-tuning the purification model with adversarial loss for robust optimization [22].

Differen from these works, we explore from an entirely new perspective to investigate how adversarial perturbation distorts the semantic relationship of image patches in Mask Autoencoder (MAE) [15]. The core idea of MAE is to randomly occlude patches of the input image with a certain ratio and recover the occluded pixels from the remaining ones. To design a robust purification method, we first identify an intriguing phenomenon of MAE. Specifically, in the case of adversarial examples subjected to tiny, visually imperceptible perturbations, the reconstruction performance of MAE is severely compromised, dealing a devastating blow. As a typical example shown in Figure 1a and1d, although the clean example and adversarial example appear very similar, MAE's reconstruction outputs in Figures 1c and 1e exhibit significant differences. The reconstruction of perturbed data, as illustrated in Figure 1e, still displays poor quality. These findings suggest that the reconstruction capability of MAE is highly sensitive to adversarial perturbations, although these perturbations are visually imperceptible. Motivated by this observation, we consider preserving the semantic relationships among image patches as a novel mechanism for adversarial purification, a direction that has not been fully explored in existing works.

Based on such a research motivation, our proposed study aims to fill the gap. In this paper, through a series of analyses, we conjecture that this phenomenon results from adversarial perturbations could easily distort semantic relations within patches, i.e. causing variation in the attention matrix (AMV), which leads to degraded image reconstruction quality of MAE. Through rigorous theoretical derivations and empirical experiments, we provide compelling evidence of sensitivity of MAE to this AMV. Concretely, as the patch attention matrix essentially reflects how different semantic patches may be related to the masked patch, altering the attention matrix means a semantic change when the masked patch is reconstructed. As Figures 2b and 2c show, the reconstruction of the target patch in the red square depends on the similar patches in the clean image, while for adversarial images high importance is assigned to distant and dissimilar patches. This suggests that the adversarial perturbations alters the semantic relations among patches. Moreover, our findings reveal that the reconstruction loss of adversarial examples is lower-bounded by the sum of the loss for clean examples and the AMV. Drawing inspiration from this finding, we propose a novel MAE-Pure method which purifies adversarial perturbations by minimizing AMV, ultimately resulting in a inter-patch semantic preserving framework.

MAE-Pure leverages the inherent sensitivity of MAE to adversarial noise, thereby achieving enhanced robustness. To further bolster defense capabilities, we demonstrate that our AMV-guided purification

is a scalable mechanism that readily adapts to more powerful generative models, such as MaskDiT [50]—a diffusion model that, while sharing architectural similarities with MAE, exhibits superior generative capabilities. Furthermore, based on the insight of the previous work [22, 48], we propose a Robust MAE-Pure (RMAE-Pure) and Robust MaskDiT-Pure (RMaskDiT-Pure) that leverages classification loss to fine-tune the purification model, significantly improving its inter-patch semantic preservation capability. We have extensively evaluated our method by comparing the important adversarial training and adversarial purification methods on various challenging adaptive attack benchmarks. Our method achieves state-of-the-art (SOTA) performance on four datasets, e.g., CIFAR-10 [18], CIFAR-100 [18], SVHN [28], and ImageNet [11].

In summary, our main contributions are as follows:

1) We investigate the susceptibility of MAE to noise interference from both theoretical and empirical perspectives and disclose that the noise induces the deviation of semantic relations among patches, resulting in a degradation of the quality of the reconstruction. On the basis of our findings, we devise a novel and efficient purification technique, called MAE-Pure, which is theoretically proven by rigorous analysis.

2) We successfully apply our insight to the mask diffusion model, e.g., MaskDiT, further enhancing model performance. Meanwhile, we further propose RMAE-Pure, which incorporates classification loss to fine-tune the purification model, significantly improving standard and robust accuracy.

3) Extensive experiments have been conducted to validate the effectiveness of our MAE-Pure on various benchmarks, showing that our approach consistently achieves favorable outcomes after denoising processes.

## 2 Preliminaries and Related Work

Due to space constraints, we have included the related work on adversarial training and MAE in Appendix E.

### 2.1 Adversarial Purification

**Adversarial Purification:** Generative models have shown great promise in purifying adversarial examples, drawing significant attention in robustness research. The early milestone study **(author?)** [32] introduced Defense-GAN, using GANs for purification. Song et al. [36] proposed the PixelDefense method, which employs the autoregressive models to mitigate the perturbations. Score-based generative models have also been applied for defense [43]. Leveraging diffusion models, DiffPure [29] uses Stochastic Differential Equation (SDE) diffusion [37] for the denoising procedure, achieving robustness. Recent works [21, 24] further improve robustness by fine-tuning diffusion models. Lin et al. [22] proposed a hybrid approach combining adversarial training with purification. It is significant to reconstruct the data without semantic information changes. Thus, Bai et al. [3] introduced contrastive guidance in diffusion models to enhance purification while preserving semantics. The adversarial purification method can be combined with other machine learning paradigms. For example, the framework of Self-supervised Online Adversarial Purification (SOAP) [34] achieves notable results by integrating self-supervised tasks during training, further boosting robustness.

Several studies have adapted MAE for denoising to enhance adversarial robustness [51, 44]. The defensive denoising model based on information discarding and robust representation restoration (DIR) [51] jointly trains a classifier and MAE with adversarial training, enabling the MAE to restore robust features by leveraging the unmasked patches to mitigate adversarial noise. Inspired by the working flow of denoising autoencoders,

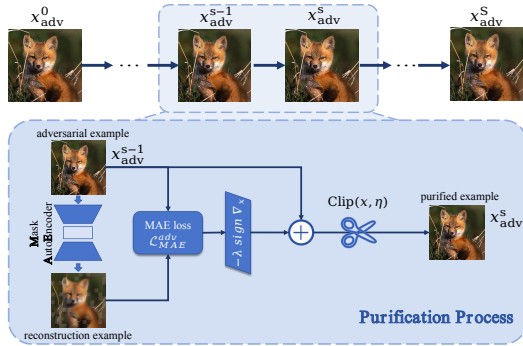

Figure 3: Overview of the proposed MAE-Pure.

both DMAE [42] and NIM-MAE [44] integrate Gaussian noise into the masked image modeling mechanism, where the attached noises will be removed during the encoding and decoding process. The framework of DMAE [42] aims to achieve a robust pre-traning process for enhancing the generalization ability and robustness over Gaussian noise without the degradation of effectiveness against adversarial attacks. NIM-MAE [44] applies the pre-trained model to remove adversarial perturbations, however, there still exists improvement space for robustness performance. Unlike previous methods, MAE-Pure leverages the sensitivity of semantic patch relationships to adversarial perturbations and employs optimization-based denoising to reduce AMV, effectively minimizing semantic variations in adversarial examples. This novel perspective has not been studied in previous research.

# 3 Theoretical Analysis

In this section, we initiate a theoretical analysis to examine the effect of adversarial perturbation on semantic relationships among patches. We will elucidate the correlation between the variation in AMV and the reconstruction loss of the MAE decoder, both theoretically and empirically.

## 3.1 Adversarial Perturbation Induces Attention Matrix Variation in MAE

Given a clean sample $x$ and its adversarial counterpart $x_{adv}$, the attention matrices and input features at the $t$-th layer in MAE are $\mathbf{A}^t$ and $\mathbf{Z}^t$ for $x$, and $\mathbf{A}^t_{adv}$ and $\mathbf{Z}^t_{adv}$ for $x_{adv}$, respectively. To save the space, more defination can be founded in Appendix E.2. The attention matrix variation (AMV) at layer $t$ induced by adversarial perturbation is formally defined as $\mathbf{A}^t_{adv} - \mathbf{A}^t$, where $\mathbf{W}^t_K, \mathbf{W}^t_Q \in \mathbb{R}^{n_t \times d_t}$ are the weight matrices at the $t$-th layer, and $N$ represents the number of training samples. AMV indicates a shift in MAE's focal points on the image, reflecting a change in the inter-patch semantic information being captured, as $\mathbf{A}^t_{adv}$ misaligns attention toward irrelevant regions and distorts the overall interpretative context (see Figure 2). To quantify how such noise affects the attention matrix, we derive Theorem 3.1 to formally express the impact of perturbation $\delta_t$ on AMV, revealing the inherent AMV sensitivity of the MAE.

**Theorem 3.1.** *Let $\delta_t = \mathbf{Z}^t_{adv} - \mathbf{Z}^t$ denotes the latent feature shift caused by the adversarial perturbation at layer $t$ in MAE. With a set $\{\omega_i\}_{i=0}^k$ and kernel coefficient $\omega_i \in \mathcal{N}(0, \mathbf{I}_d)$, it holds that:*

$$||\mathbf{A}^t_{adv} - \mathbf{A}^t||_2 \approx \gamma \left\| \left[ \left( \mathbf{Y} - \mathbf{B}\mathbf{Q}^t \right)^\top \mathbf{W}^t_Q + \left( \mathbf{Y} - \mathbf{B}\mathbf{K}^t \right)^\top \mathbf{W}^t_K \right] \delta_t \right\|_2,$$

$$where \quad \mathbf{B} = \sum_{i=0}^k \exp\left( \omega_i^\top (\mathbf{Q}^t + \mathbf{K}^t) \right), \quad \mathbf{Y} = \sum_{i=0}^k \exp\left( \omega_i^\top (\mathbf{Q}^t + \mathbf{K}^t) \right) \omega_i, \quad \gamma = \frac{\exp\left( -\frac{\|\mathbf{Q}^t\|^2 + \|\mathbf{K}^t\|^2}{2} \right)}{m}.$$

*Proof.* The proof can be seen in Appendix K.2. $\qquad\square$

Theorem 3.1 suggests that even minor shifts in the latent features ($\delta_t$) may have the ability to cause disproportionately large changes in AMV, especially due to the high dimensionality $d_t$ of internal projection matrices. This aligns with prior findings [17] showing that transformers are inherently sensitive to input perturbations. Notably, in MAE, where $d_t \gg$ input dimension, the sensitivity is further amplified. This analysis reveals the intrinsic vulnerability of MAE's attention mechanism under adversarial conditions, offering a theoretical foundation for the empirical trends shown in Figure 4 (a-c) of Section 3.3.

## 3.2 Impact of Decoder Attention Shifts on Adversarial Reconstruction in MAE

To deepen the understanding of how attention pattern distortions affect output quality in MAE, we present a theoretical lower bound on the reconstruction loss under adversarial conditions. This analysis extends the discussion of AMV sensitivity in Section 3.1 and reveals how attention shifts in the decoder layer affect reconstruction loss. First, let $\mathcal{L}^{adv}_{rec}$ denote the average reconstruction loss for adversarial examples, corresponding to the reconstruction loss $\mathcal{L}_{rec}$ for clean samples as Eq. (4) in Appendix E.2.

**Theorem 3.2.** *Let $\mathbf{A}_{i,t}^{\mathrm{dec}}$ denote the attention matrix at the $t$-th layer of the MAE decoder for the $i$-th sample in the dataset, and let $\mathbf{A}_{\mathrm{adv},i,t}^{\mathrm{dec}}$ denote the corresponding attention matrix for the adversarial examples. With ratio constants $C_A$ and $H$, it holds that:*

$$\mathcal{L}_{rec}^{\mathrm{adv}} \geq \frac{1}{2}\mathcal{L}_{rec} + \frac{1}{2NT} \sum_{t=1}^{T} \sum_{i=1}^{N} \left[ HC_A \left\| \mathbf{A}_{\mathrm{adv},i,t}^{\mathrm{dec}} - \mathbf{A}_{i,t}^{\mathrm{dec}} \right\|^2 - c_{rec} \right],$$

*where $c_{rec}$ is the reconstruction bias, which symbolizes the disparity between the output of MAE and the original, unmasked image. The definition of $c_{rec}$ can be found in Appendix K.1.*

*Proof.* The proof can be seen in the Appendix K.3. □

It shows that the lower bound of the reconstruction loss for adversarial data can be decomposed into three components: the average reconstruction loss for clean data $\mathcal{L}_{\mathrm{rec}}$, the average attention matrix variation for the MAE decoder at each layer $\frac{1}{2NT} \sum_{t=1}^{T} \sum_{i=1}^{N} ||(A_{adv_{i,t}}^{dec} - A_{i,t}^{dec})||^2$, and constant terms. Theorem 3.2 demonstrates that adversarial distortions in AMV of decoder lead to increases in reconstruction loss $\mathcal{L}_{\mathrm{rec}}$, and Figure 4 provides further evidence of this phenomenon. Notably, the reconstruction loss is shown to be consistent with the degree of AMV, confirming a strong correlation between inter-patch semantic relationships and output degradation. This theorem builds a theoretical foundation of robust MAE-based purification.

## 3.3 Empirical Validation

This section describes the empirical validation of our theoretical analysis. First, we describe the impact of MAE reconstruction on adversarial perturbations. We then examine the correlation between the changes in semantic relations within patches (attention matrix variation) and the reconstruction loss. Finally, we present the empirical evidence that adversarial perturbations affect the semantic relationships within MAE patches. Meanwhile, we also verify the sensitivity of AMV to adversarial perturbations.

**Perturbation Leads to Degraded Reconstruction Quality.** To empirically investigate how perturbation influences reconstruction, an image is randomly selected from the SVHN dataset. We then compare the reconstruction results of the clean data and the adversarial example as shown in Figure 1. The adversarial example generated by the AutoAttack procedure [10] in Figure 1d looks almost identical to its clean counterpart in Figure 1a visually. However, its reconstruction (Figure 1e) is significantly different from that of the clean sample (Figure 1c). Likewise, its reconstruction result also shows significant differences with a reconstruction of its clean sample (Figure 1c). These phenomena emphasize the substantial impact of adversarial perturbations on the overall outcome of the reconstruction.

**Visualization of Attention Matrix Variation.** To check how the semantic relationship between different patches changes under perturbation, we show the degree of importance of visible patches for reconstructing the masked patch in Figure 2. We select an image from ImageNet [11] and randomly generate a mask matrix. A specific masked patch is considered as the target patch (marked red in a box) in Figure 2a. Then, the visible patches are fed into the MAE to perform the reconstruction. The degree of importance of visible patches for reconstructing the target patch, which is determined by the corresponding values of the attention matrix, is illustrated in Figure 2b (e.g., the last layer attention of the MAE decoder). Meanwhile, we also display the corresponding visualizations of the adversarial example and the denoised example for the same mask matrix and target patch position.

In generating attention weights for the target patch (red box), the approach involves using the patch itself as the query vector and the remaining patches as key vectors. Through self-attention, the attention weights are determined. Higher weights indicate a stronger semantic similarity between the target patch and the patch itself. Figures 2b, 2c, and 2d show the importance degree of visible patches in the clean sample, adversarial example, and denoised example, respectively. Patches that are closer to red are more significant. As demonstrated in Figure 2b, when a hole in the tape is used as the target patch, the visible patch of another similar hole and the surrounding patches serve as the most important basis for reconstructing the target patch. However, as illustrated in Figure 2c, some distant and irrelevant visible patches with large color and shape differences are taken or focused to

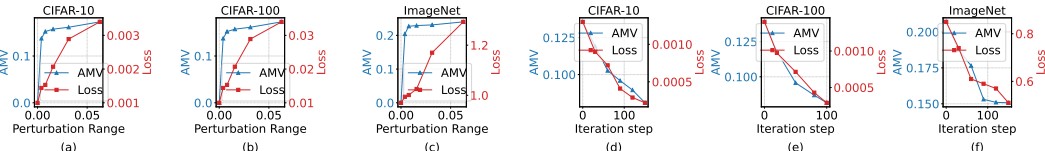

Figure 4: Trends of MAE reconstruction loss and attention matrix variation under AutoAttack and during purification across multiple datasets. (a–c): Under AutoAttack on CIFAR-10, CIFAR-100, and ImageNet with different attack budget. (d–f): During the purification process with MAE-Pure on CIFAR-10, CIFAR-100, and ImageNet.

reconstruct the adversarial perturbed target patch. This indicates that the adversarial perturbation leads MAE to the erroneous attention.

In addition, we substantiate our point of view by presenting additional examples. In Figure 8b (Supplement G), the target, which includes hands and the instrument, naturally takes into account the other hands and the patch of the instrument for reconstruction. However, when perturbed by adversarial noises, MAE deems patches in the distant background as more important in Figure 8c. Similarly, in Figure 8f, patches of a girl's eye, located within the patch of the other eye, are considered more important for reconstruction, but adversarial perturbation leads MAE to prioritize the ground area near the border in Figure 8g. After applying MAE-Pure, we observe that the denoised image shows a judgment of the importance of other patches during target reconstruction that closely approximates the clean sample, as illustrated in Figure 8d and 8h.

**Analysis of AMV Sensitivity and Consistency with Reconstruction Loss.** To validate Theorem 3.2 and the effect of attention matrix variation on the reconstruction quality, we visualize the changes of reconstruction loss and attention matrix variation with respect to the intensity of adversarial noise in Figure 4. To evaluate the average reconstruction loss and average AMV $\frac{1}{N} \sum_{i=1}^{N} ||\mathbf{A}_{adv,i}^{dec} - \mathbf{A}_i^{dec}||_2$ for adversarial examples, we randomly selected 150 images from CIFAR-10, CIFAR-100, and ImageNet, and applied a mask ratio of 0.5. We exploit the AutoAttack method to generate adversarial examples. Clearly, AMV is highly sensitive to perturbations, with its values rising rapidly even with minimal changes (e.g., $\delta = 0.01$) in Figure 4 (a-c). Notably, AMV continues to increase steadily until it approaches its upper limit. Since each value in the attention matrix is bounded (due to the effect of softmax), AMV also has an upper limit. As perturbations increase, AMV gradually approaches this limit, causing the increase to level off. In addition, the reconstruction loss and AMV share a consistent trend as the level of perturbation increases. This is consistent with our theoretical analysis of the relationship between AMV and MAE reconstruction loss, as well as the sensitivity of AMV to perturbations.

# 4 Method

## 4.1 Adversarial Purification with MAE-Pure

As discussed in Section 3, the MAE attention mechanism is highly sensitive to adversarial perturbations, which distort inter-patch semantic relationships and degrade reconstruction quality. Based on this sensitivity, we propose a purification scheme, MAE-Pure, which formulates denoising as an optimization problem that minimizes semantic variations.

We denote the clean data as $x$, and the adversarial example as $x_{adv}$. In the context of denoising, the objective is to induce a modification $\Delta$ on $x_{adv}$ such that the attention matrix of the denoised examples, $atten(x_{adv} + \Delta)$, closely aligns with the attention matrix of clean data samples $atten(x)$. Therefore, the learning objective of denoising can be formed as an Attention Matrix Variation Minimization problem (AMVM), which is denoted as:

$$\min_{\Delta} \mathcal{L}(\Delta) = ||\text{atten}(x_{adv} + \Delta) - \text{atten}(x)||_2,$$
$$\text{s.t. } ||\Delta||_\infty \leq C_e, \tag{1}$$

where $C_e$ is a small constant.

In addressing the AMVM problem, the denoising process strives to mitigate adversarial perturbations on $x_{adv}$. Its goal is to achieve a consistent attention matrix between the denoised images and the clean images of MAE. This alignment ensures that the patch relationship of the denoised image closely

approximates that of clean samples, thereby minimizing the impact of adversarial perturbations in the denoising outcome.

Since clean attention $\text{atten}(x)$ is unavailable during inference, directly minimizing AMV Eq. (1) is intractable. Instead, inspired by Theorem 3.2, we minimize the MAE reconstruction loss as a tractable surrogate. Reconstruction diffusion loss not only enables efficient optimization without requiring clean attention (tractability), but also can reduce AMV caused by perturbations (owing to the consistent trend presented in Theorem 3.2), thereby aligning with the inter-patch semantic structure of the clean input. As such, our approach instead minimizes the MAE reconstruction loss for adversarial purification.

$$min_\Delta \mathcal{L}_{\text{rec}}(x_{adv} + \Delta) \iff min_\Delta \mathcal{L}(\Delta),$$
$$s.t. \quad ||\Delta||_\infty \leq C_e. \tag{2}$$

To address this problem, we employ the standard Projected Gradient Descent (PGD) method [26]. In this approach, the modifications are iteratively added to adversarial examples, and the total number of iterations is denoted as $S$. At the $s$-th iteration, the denoising process is denoted as:

$$x_{adv}^s = Clip(x_{adv}^{s-1} - \lambda \cdot \Delta_s, \eta),$$
$$\Delta_s = \text{sign}(\nabla_x L_{\text{rec}}(x_{adv}^{s-1})). \tag{3}$$

Here $\mathcal{L}_{\text{rec}}$ signifies the MAE reconstruction loss defined in Eq. (4), with $\lambda$ representing the step size and $\eta$ as the clipping threshold. The overall modification $\Delta$ is composed of individual iteration modification $\Delta_s$ for $s \in [1, S]$. The purpose of $\Delta$ is to guide the attention distribution of adversarial examples $\text{atten}(x_{adv})$ towards the clean sample distribution $\text{atten}(x)$. The denoising algorithm and pipeline of our MAE-Pure can be seen in Algorithm 1 (Supplement E) and the whole process is in Figure 3.

Table 1: Clean and robust accuracy (%) on CIFAR-10 obtained by different purification methods.

| Method | Architecture | Std Acc | Robust Acc | |
|---|---|---|---|---|
| | | | $\ell_\infty$ | $\ell_2$ |
| Shi et al. [34] | WideResNet-28-10 | 91.89 | 4.56 | 7.25 |
| Yoon et al. [43] | WideResNet-70-16 | 87.93 | 37.65 | 57.81 |
| Zhang et al. [46] | WideResNet-70-16 | **93.16** | 22.07 | 35.74 |
| Diffpure [29] | WideResNet-70-16 | 92.50 | 42.20 | 60.80 |
| COUP [48] | WideResNet-28-10 | 90.33 | 41.72 | 57.25 |
| ADBM [21] | WideResNet-70-16 | 91.90 | 47.70 | 63.30 |
| ADDT$_{w/\text{Diffpure}}$ [24] | WideResNet-28-10 | 89.94 | 55.76 | - |
| MAE-Pure | WideResNet-28-10 | 88.57 | 40.53 | 53.50 |
| MaskDiT-Pure | WideResNet-28-10 | 92.03 | 50.57 | 64.53 |
| RMAE-Pure | WideResNet-28-10 | 90.09 | 45.15 | 60.72 |
| RMaskDiT-Pure | WideResNet-28-10 | 93.11 | **62.13** | **73.57** |

To empirically validate the theory of MAE-Pure, we also plot the trends of the MAE recontrusction loss and AMV with increasing denoising iterations in Figure 4 (d-f). We randomly selected 100 adversarial examples from each dataset, perturbed using AutoAttack. For CIFAR10 and CIFAR100, the perturbation magnitude is set to $\ell_\infty = \frac{8}{255}$, while for ImageNet, it is set to $\ell_\infty = \frac{4}{255}$. As observed, both the AMV and loss exhibit a similar downward trend as the number of purification iterations increases. This supports the validity of our theory.

Furthermore, we provide strict convergence analysis within the Appendix section J.

## 4.2 Extension to Mask Diffusion Model

The strong sensitivity of MAE to adversarial perturbations reveals a key vulnerability in masked image modeling: inter-patch semantic information (AMV) is sensitive to noise. Based on the high sensitivity of AMV to adversarial perturbations, we designed a more robust method, MAE-Pure. Furthermore, we aim to combine the sensitivity of AMV with more powerful generative models (e.g., diffusion) to develop a stronger and more effective adversarial denoising approach. To this end, we consider MaskDiT [50], a diffusion-based extension of MAE that preserves its masked autoencoding architecture while integrating forward and reverse diffusion steps for high quilty image generation. Given their architectural alignment, we posit that MAE's observations—such as perturbation sensitivity, AMV, and reconstruction dynamics—can be transferred to MaskDiT. Empirical comparisons support this view: as shown in Figure 7 of Appendix G, MaskDiT exhibits consistent patterns with MAE under adversarial settings. Motivated by this phenomenon, we propose MaskDiT-Pure, which leverages MaskDiT's generative capacity while retaining MAE's inter-patch semantic sensitivity, further improving purification performance by optimizing its reconstruction loss (Eq. (2)).

## 4.3 Robust Purification Model

As the study of AToP [22] shows, further fine-tuning a purification model using classification loss can enhance its robustness against both seen and unseen attacks. Following this insight, we propose a two-stage fine-tuning method to develop Robust MAE-Pure (RMAE-Pure) and Robust MaskDiT-Pure (RMaskDiT-Pure) variants to enhance the semantic relationship-preserving capabilities. For more details about our method, refer to Appendix F.1.

Figure 6a (see Appendix F.1) shows a quantitative analysis of enhanced semantic relationships using AMV as an evaluation metric. We randomly select 100 images from the CIFAR-10 dataset and examined the AMV of MAE-Pure and RMAE-Pure under the AutoAttack with an attack budget of $\frac{8}{255}$ across different purification iterations. Specifically, the initial AMV (without purification) of RMAE is higher than that of MAE-Pure. This suggests that RMAE is more sensitive to interpatch semantic information changes caused by adversarial attacks. However, with the progression of purification iterations, the AMV of RMAE-Pure decreases significantly, highlighting its superior capability in preserving semantic integrity compared to MAE-Pure. Furthermore, as shown in Figure 6b (Appendix F.1), both MaskDiT-Pure and RMaskDiT-Pure exhibit similar trends.

# 5 Experiment

## 5.1 Experimental Setting

**Datasets and Classifier.** In this section, we validate the robustness of our purification method, MAE-Pure, on four benchmark datasets, including CIFAR-10 [18], CIFAR-100 [18], SVHN [28], and ImageNet [11]. We use WideResNet-28-10 [45] as the main classifier for CIFAR-10, CIFAR-100, and SVHN, and ResNet-101 [16] as the main classifier for ImageNet.

**Adversarial Attacks.** Several studies [6, 21, 24] show that the AutoAttack method [10] tends to overestimate the robustness of diffusion models, primarily due to the presence of gradient obfuscation, which prevents the attack from effectively exploiting the true vulnerabilities of the model. To address this issue, recent studies [6, 21, 24] have adopted the gradient checkpointing technique to efficiently extract complete gradients throughout the diffusion process. Furthermore, Li et al. [21] have fur-

Table 2: Clean and robust accuracy (%) on CIFAR-100 and SVHN obtained by different purification methods. The experiment are implemented on WideResNet-28-10.

| Dataset | CIFAR100 | | | SVHN | | |
|---|---|---|---|---|---|---|
| Method | Std Acc | Robust Acc | | Std Acc | Robust Acc | |
| | | $\ell_\infty$ | $\ell_2$ | | $\ell_\infty$ | $\ell_2$ |
| Diffpure [29] | 45.23 | 11.57 | 31.53 | 93.90 | 39.70 | 63.30 |
| COUP [48] | 65.71 | 15.22 | 34.28 | 92.07 | 41.62 | 63.97 |
| ADDT$_w$/DDPM [24] | 66.02 | 18.85 | 36.57 | - | - | - |
| ADBM [21] | - | - | - | 93.50 | 47.90 | 65.70 |
| MAE-Pure | 65.34 | 14.28 | 29.29 | 94.54 | 27.59 | 55.29 |
| MaskDiT-Pure | **70.03** | 24.39 | 36.51 | 94.91 | 46.57 | 66.38 |
| RMAE-Pure | 66.28 | 19.53 | 31.58 | 94.47 | 39.15 | 60.51 |
| RMaskDiT-Pure | 69.87 | **29.91** | **43.27** | **95.39** | **55.90** | **70.18** |

ther demonstrated that, compared to AutoAttack, the combination of PGD + EOT is more effective in evaluating the adaptive defense mechanisms of diffusion models. In line with these studies [21], we employ the PGD200 + EOT20 configuration with $\ell_\infty(\epsilon = \frac{8}{255})$ and $\ell_2(\epsilon = 1)$, utilizing the exact gradient computation method described in [21, 24] to ensure a more robust evaluation of defense performance in our experiments. Moreover, to ensure a fair comparison with adversarial training methods, we adopt AutoAttack with full gradient settings [6] as the evaluation protocol, thereby guaranteeing the objectivity and comparability of the results.

**Evaluation Metrics.** To evaluate the model's performance, we employ two metrics for classification: robust accuracy (**Robust Acc**) and standard accuracy (**Std Acc**), which are tested respectively on adversarial examples and clean samples. Due to the high computational cost of testing models with multiple attacks, we follow previous work [29, 22, 21] and randomly select 512 test samples from each testing dataset.

Table 3: Comparison with adversarial training under Autoattack ($\epsilon = \frac{8}{255}$)

| Method | Extra data | Architecture | CIFAR10 | | CIFAR100 | | SVHN | |
|---|---|---|---|---|---|---|---|---|
| | | | Std Acc | $\ell_\infty$ | Std Acc | $\ell_\infty$ | Std Acc | $\ell_\infty$ |
| Rebuffi et al.[31] | ✓ | WRN-28-10 | 87.33 | 60.73 | 62.41 | 32.06 | 94.34 | 60.90 |
| Pang et al. [30] | ✓ | WRN-28-10 | 88.10 | 61.51 | 62.08 | 31.40 | – | – |
| Wang et al.[40] | ✓ | WRN-28-10 | 91.12 | 63.35 | 68.06 | 35.65 | 95.19 | 61.85 |
| MAE-Pure | ✗ | WRN-28-10 | 88.57 | 40.65 | 65.34 | 16.77 | 94.54 | 47.03 |
| MaskDiT-Pure | ✗ | WRN-28-10 | 92.03 | 64.97 | 70.03 | 33.51 | 94.91 | 64.15 |
| RMAE-Pure | ✗ | WRN-28-10 | 90.09 | 47.15 | 66.28 | 22.15 | 94.47 | 50.03 |
| RMaskDiT-Pure | ✗ | WRN-28-10 | **93.11** | **75.83** | **69.87** | **40.13** | **95.39** | **66.12** |

## 5.2 Compare with the State-of-the-art

We compare our results with state-of-the-art methods across four datasets: CIFAR-10, CIFAR-100, SVHN, and ImageNet. **Due to the space limitations, we provide detailed comparisons under the PGD200 + EoT20 attack for CIFAR-10, CIFAR-100, and SVHN in main paper. More experimental results, including performance on ImageNet, transferability of the fine-tuned purification model, defense against extra attacks, time-consumption analysis, ablation studies, sensitivity analysis, and evaluations between different classifiers, are also presented in A.**

**CIFAR-10.** Table 1 highlights the performance of various purification methods on the CIFAR-10 dataset in terms of Std Acc and Robust Acc. RMaskDiT-Pure excels with 62. 13% robust accuracy in $\ell_\infty$ attacks, 73. 57% in $\ell_2$ attacks, and strong standard accuracy of 93. 11%. In contrast, traditional methods like the previous method [34] perform poorly, achieving only 4.56% under $\ell_\infty$ attacks. In general, our method significantly enhances adversarial robustness. In addition, our experiments utilizing the WideResNet-70-16 architecture are presented in Supplementary Material B.6.

**CIFAR100.** Table 2 summarizes the performance of various purification methods on CIFAR-100. Our proposed method achieves strong performance, with MaskDiT-Pure reaching the highest standard accuracy of 70.03% and a robust accuracy of 24.39% under $\ell_\infty$ attacks and 36.51% under $\ell_2$ attacks. RMaskDiT-Pure further improves robustness, achieving the best $\ell_\infty$ robust accuracy of 29.91% and $\ell_2$ robust accuracy of 43.27%, outperforming other methods like Diffpure and COUP.

**SVHN.** Table 2 shows that ADBM achieves strong robust accuracy (47.90% under $\ell_\infty$ attacks and 65.70% under $\ell_2$) attacks but is outperformed by RMaskDiT-Pure, which achieves the best robust accuracy (55.90% and 70.18%) with comparable standard accuracy. While RMAE-Pure leads in standard accuracy (95.39%), its robustness is lower. Overall, our MaskDiT-based methods achieve a better balance between clean and robust performance compared to ADBM.

## 5.3 Comparison with adversarial training

As shown in Table 3, our methods achieve competitive or superior robustness compared to adversarial training baselines, **without using any extra data**. In contrast, prior works [31, 30, 40] rely on **1M additional training samples**. Notably, **RMaskDiT-Pure achieves 75.83% robust accuracy on CIFAR-10**, outperforming all baselines and highlighting the effectiveness of our data-free approach.

## 6 Conclusion

This paper reveals the vulnerability of MAEs to subtle adversarial attacks, caused by adversarial noises that disrupt semantic relations in image patches. To address this, we propose the MAE-Pure pipeline, a denoising method using Attention Matrix Variation Minimization, which iteratively refines adversarial examples by minimizing reconstruction loss to converge to clean images. We further enhance MAE-Pure with classifier loss, introducing RMAE-Pure, and extend it to diffusion models with MaskDiT-Pure. Extensive experiments demonstrate that our methods achieve state-of-the-art performance on multiple benchmarks.

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

## A  Limitation

One limitation of MAE-Pure lies in its linear memory consumption with respect to batch size, which may restrict scalability under limited GPU resources.

## B  Broader impact

We are the first to systematically explore the relationship between adversarial noise and inter-patch semantic information. While existing defense methods primarily focus on suppressing pixel-level perturbations or enhancing model robustness structurally, our work takes a novel perspective by analyzing reconstruction consistency and semantic alignment. We reveal how adversarial perturbations disrupt inter-patch semantic relations and propose a reconstruction paradigm that restores this consistency. This new angle provides a valuable direction for future adversarial defense research and advances the theoretical and practical understanding of robustness from a structure-aware perspective.

## C  Code of this work

The code of this article is available at: `https://anonymous.4open.science/r/MAE-Pure-7F85`
.

## D  Supplement Experiment

We have enhanced this section with additional experiments to provide a more comprehensive evaluation of our work. Specifically, we present ImageNet [11] results under PGD200 + EoT20 [26, 2] with the perturbation budgets $\epsilon = \frac{4}{255}$ for $\ell_\infty$ attack and $\epsilon = 0.5$ for $\ell_2$ attack, using ResNet-101 [16] as the classifier. We also performed ablation studies on CIFAR-10 [18], CIFAR-100 [18], and SVHN [28] using different backbones and evaluated diverse attack scenarios.

### D.1  Performance on ImageNet.

Table 4 presents the standard accuracy and robust accuracy of different purification methods on the ImageNet dataset under the $\ell_\infty$ attack. As shown in the table, the ADDT method achieves the highest standard accuracy at **80.20%**, slightly outperforming the other methods. However, in terms of robustness, RMaskDiT-Pure stands out with a robust accuracy of **36.87%**, surpassing all other methods, including ADDT. In contrast, MAE-Pure and MaskDiT-Pure demonstrate relatively lower robustness, achieving 24.75% and 32.29%, respectively.

Table 4: Clean and robust accuracy (%) with $\ell_\infty$ and $\ell_2$ attack on ImageNet obtained by different purification methods.

| Defense model | Std Acc | Robust Acc | |
|---|---|---|---|
| | | $\ell_\infty$ | $\ell_2$ |
| Diffpure [29] | 77.51 | 30.15 | 44.15 |
| ADDT [23] | **80.20** | 35.83 | - |
| MAE-Pure | 67.53 | 24.75 | 35.95 |
| MaskDiT-Pure | 75.52 | 32.29 | 45.57 |
| RMAE-Pure | 78.85 | 29.48 | 42.25 |
| RMaskDiT-Pure | 79.52 | **36.87** | **51.17** |

### D.2  Transferability of finetuned purification on new classifiers

We fine-tune the RMAE-Pure/RMaskDiT-Pure model based on WideResNet-28-10 [45] and replace it with different classifiers for testing experiments. The WideResNet-70-16 [45] and ResNet-50 [16] are selected for testing process to observe the transferability of our proposed method across different classifiers.

Table 5: Robust accuracy (%) of different purification methods against $\ell_\infty(\epsilon = \frac{8}{255})$ and $\ell_2(\epsilon = 1)$ adversarial attacks across two classifiers: WideResNet-70-16 and ResNet-50. Here, our method are derived by fine-tuning on the WideResNet-28-10 classifier.

| Classifier | WideResNet-70-16 | | ResNet-50 | |
|---|---|---|---|---|
| | $\ell_\infty$ | $\ell_2$ | $\ell_\infty$ | $\ell_2$ |
| Diffpure [29] | 42.20 | 60.80 | 38.02 | 54.74 |
| RMAE-Pure | 44.27 | 62.42 | 43.72 | 60.08 |
| RMaskDiT-Pure | **58.37** | **68.55** | **54.22** | **67.15** |

The results in Table 5 highlight the superior performance of our method, particularly RMaskDiT-Pure, which achieves the highest robust accuracy across both classifiers and attack norms. Notably, the fine-tuned models, initially trained on WideResNet-28-10, demonstrate strong transferability when applied to WideResNet-70-16 and ResNet-50 without the need for retraining. This finding underscores the practicality and scalability of our method, as its fine-tuned models can be seamlessly adapted to new classifiers, providing an efficient and robust defense against adversarial attacks.

## D.3 Performance on unseen threats

Table 6: Robust accuracy (%) against unseen threats with the setting of $\ell_1(\epsilon = 12)$ and $\ell_2(\epsilon = 1)$.

| Defense model | CIFAR-10 | | CIFAR-100 | | SVHN | |
|---|---|---|---|---|---|---|
| | $\ell_1$ | $\ell_2$ | $\ell_1$ | $\ell_2$ | $\ell_1$ | $\ell_2$ |
| Diffpure [29] | 44.30 | 60.80 | 13.51 | 27.53 | 46.10 | 63.30 |
| ADBM [21] | 49.60 | 63.30 | - | - | 51.20 | 65.70 |
| RMAE-Pure | 44.41 | 60.72 | 12.97 | 29.58 | 47.09 | 60.51 |
| RMaskDiT-Pure | **65.11** | **73.57** | **41.15** | **43.27** | **55.53** | **70.18** |

For the three methods, ADBM, RMAE-Pure, and RMaskDiT-Pure, all of which are fine-tuned under the $\ell_\infty$ norm, the $\ell_1$ and $\ell_2$ norms are considered as unseen threats. To verify the robustness of the proposed method, we will now conduct testing under these unseen threats. For a fair comparison on the CIFAR-10 dataset, we employ the WideResNet-70-16 architecture, and we use the WideResNet-28-10 architecture on the CIFAR-100 and SVHN datasets.

Table 6 presents the robust accuracy (%) of different defense models against unseen threats ($\ell_1$ and $\ell_2$ attacks) on the CIFAR-10, CIFAR-100, and SVHN datasets. As a baseline method, DiffPure [29] performs moderately on CIFAR-10 and SVHN but poorly on CIFAR-100, especially under $\ell_1$ attacks (13.51%). ADBM outperforms DiffPure on CIFAR-10 and SVHN, but no data is provided for CIFAR-100, suggesting potential limitations or untested performance on this dataset. Our RMAE-Pure method slightly outperforms DiffPure on CIFAR-10 and SVHN but underperforms on CIFAR-100 (12.97% vs. 13.51%), indicating some limitations on more complex datasets. In contrast, RMaskDiT-Pure significantly outperforms all other methods across all datasets and attack types. On CIFAR-10, RMaskDiT-Pure achieves accuracies of 65.11% and 73.57% under $\ell_1$ and $\ell_2$ attacks, respectively, far surpassing other methods. On CIFAR-100, although its performance under $\ell_2$ attacks is slightly lower than under $\ell_1$, it still outperforms other methods. On the SVHN dataset, RMaskDiT-Pure also demonstrates considerable robustness performance, particularly under $\ell_2$ attacks (70.18%). Overall, RMaskDiT-Pure exhibits the strongest robustness against unseen threats, especially on CIFAR-10 and CIFAR-100, showcasing its superior generalization and defense capabilities, while RMAE-Pure, though slightly less effective, still outperforms baseline methods in certain scenarios.

## D.4 Defense against adaptive attacks

Table 7 presents the robust accuracy (%) of various defense methods under different adversarial attacks in the adaptive $\ell_2(\epsilon = 1)$-norm setting on the CIFAR-10 dataset. The evaluated adaptive attacks include C&W [5]+EOT [2], DeepFool [27]+EOT, AutoAttack [10]+EOT, and PGD [26]+EOT.

Among the methods, Diffpure and ADBM represent baseline defense approaches, with ADBM generally outperforming Diffpure across all attacks. For instance, ADBM achieves 78.40% robust accuracy against C&W+EOT compared to Diffpure's 74.80%. The pure methods (non-adversarial training approaches) show varying performance: MAE-Pure exhibits the lowest robust accuracy across all attacks, while MaskDiT-Pure demonstrates stronger performance, particularly against DeepFool+EOT (82.8%). RMAE-Pure shows moderate results, and RMaskDiT-Pure consistently outperforms all other methods, achieving the highest robust accuracy against every attack, with 80.58% for C&W+EOT, 86.11% for DeepFool+EOT, 73.59% for AutoAttack+EOT, and 69.57% for PGD+EOT. This indicates that RMaskDiT-Pure is the most effective defense method in this setting, offering superior robustness across diverse adversarial attacks.

Table 7: Robust Accuracy (%) of various defense methods under different attacks in the $\ell_2(\epsilon = 1)$-norm setting using the exact gradient with WideResNet-70-16 on CIFAR-10.

| Method | C&W+EOT | DeepFool+EOT | AutoAttack+EOT | PGD+EOT |
|---|---|---|---|---|
| Diffpure[29] | 74.80 | 78.40 | 63.90 | 60.80 |
| ADBM[21] | 78.40 | 84.30 | 66.80 | 66.30 |
| MAE-Pure | 62.54 | 65.15 | 52.75 | 53.50 |
| MaskDiT-Pure | 79.15 | 82.80 | 66.43 | 64.53 |
| RMAE-Pure | 72.20 | 72.29 | 59.11 | 60.72 |
| RMaskDiT-Pure | **80.58** | **86.11** | **73.59** | **69.57** |

## D.5 Extra experiments on different classifier

Table 8 shows our standard and robust accuracy using WideResNet-70-16 under CIFAR-10 and SVHN. Compared with WideResNet-28-10, it shows better results. It means the overparameterization contributes model's robustness. Among all the methods, the effectiveness of the RMaskDiT-Pure method is most notable.

Table 8: Performance of standard accuracy and robust accuracy (%) using WideResNet-70-16.

| Method | Architecture | CIFAR10 | | | SVHN | | |
|---|---|---|---|---|---|---|---|
| | | Std Acc | $\ell_\infty$ | $\ell_2$ | Std Acc | $\ell_\infty$ | $\ell_2$ |
| MAE-Pure | WideResNet-70-16 | 89.66 | 42.21 | 56.77 | 94.97 | 17.11 | 32.75 |
| MaskDiT-Pure | WideResNet-70-16 | 94.91 | 52.07 | 66.55 | 94.93 | 46.03 | 63.77 |
| RMAE-Pure | WideResNet-70-16 | 91.07 | 47.92 | 60.95 | 94.78 | 40.79 | 60.51 |
| RMaskDiT-Pure | WideResNet-70-16 | **93.85** | **63.94** | **75.50** | **95.91** | **56.02** | **69.18** |

## D.6 Performance on Black-box Attack

To evaluate the effectiveness of against black-box attacks, we adopt three black-box attack methods: FAB [9], Square [1], and Rays [7] on CIFAR-10 and SVHN. The black-box scenario implies that the attacker has no knowledge of the defense method. Table 9 shows the robustness of various methods against black-box $\ell_\infty$ attacks with the perturbation budget $\epsilon = \frac{8}{255}$ using WideResNet-28-10. RMaskDiT still achieves the best results.

## D.7 Inference Time Comparison

Table 10 compares the inference time between different defense models in CIFAR-10 and ImageNet. We calculate the run-time for all methods with a batch size of 32, and our experiments are conducted on an A40 GPU. For CIFAR-10, **RMAE-Pure** achieves the fastest time (**11.77s**), followed by Diffpure (**12.39s**) and MAE-Pure (**18.25s**). On ImageNet, **MAE-Pure** is the most efficient (**31.51s**), significantly outperforming Diffpure (**81.54s**). The results highlight that the DiffPure model has the

Table 9: Robust accuracy (% ) against different black-box attacks $\ell_\infty(\epsilon = \frac{8}{255})$ with WideResNet-28-10. The "Vanilla" setting represents the model trained on clean datasets without any defense.

| Method | Architecture | CIFAR-10 | | | | SVHN | | | |
|--------|--------------|----------|--|--|--|------|--|--|--|
| | | Std Acc | Robust Acc | | | Std Acc | Robust Acc | | |
| | | | Square | FAB | RayS | | Square | FAB | RayS |
| Vanilla | WideResNet-28-10 | 96.75 | 19.15 | 0.00 | 1.23 | 98.11 | 9.08 | 14.78 | 16.89 |
| Diffpure [29] | WideResNet-28-10 | 89.15 | 89.15 | 88.29 | 90.51 | 93.93 | 92.15 | 93.13 | 92.97 |
| ADBM [21] | WideResNet-28-10 | - | - | - | - | 93.49 | 93.32 | 92.98 | 93.16 |
| MAE-Pure | WideResNet-28-10 | 88.57 | 78.59 | 76.43 | 77.29 | 94.54 | 92.57 | 93.36 | 93.41 |
| MaskDiT-Pure | WideResNet-28-10 | 92.03 | 90.96 | 92.25 | **93.39** | 94.91 | 92.80 | 92.59 | 92.73 |
| RMAE-Pure | WideResNet-28-10 | 90.09 | 90.25 | 89.15 | 92.31 | 94.47 | 92.73 | 93.27 | 93.36 |
| RMaskDiT-Pure | WideResNet-28-10 | **93.11** | **93.27** | **93.38** | 93.03 | **95.39** | **94.15** | **94.18** | **94.88** |

advantage of inference time for smaller datasets, while our proposed model performs better on the metrics of inference time on the ImageNet dataset.

Table 10: Inference time (s) consumption comparison across different defense models on CIFAR-10 and ImageNet datasets.

| Defense Model | CIFAR10 | ImageNet |
|---------------|---------|----------|
| Diffpure [29] | 12.39 | 81.54 |
| MAE-Pure | 18.25 | **31.51** |
| MaskDiT-Pure | 32.85 | 79.27 |
| RMAE-Pure | **11.77** | 27.38 |
| RMaskDiT-Pure | 29.73 | 62.52 |

## D.8 Robust under BPDA attack

We evaluate the robustness of our model under a strong white-box attack setting using BPDA combined with EoT set to 20. The results show as follow:

Table 11: Clean and robust accuracy (%) under BPDA attack ($\epsilon = \frac{8}{255}$) on CIFAR-10.

| Method | Architecture | Std Acc | Robust Acc ($\ell_\infty$) |
|--------|--------------|---------|----------------------------|
| Diffpure [29] | WRN-28-10 | 89.20 | 78.53 |
| MAE-Pure | WRN-28-10 | 88.57 | 78.89 |
| MaskDiT-Pure | WRN-28-10 | 92.03 | 83.44 |
| RMAE-Pure | WRN-28-10 | 90.09 | 80.17 |
| RMaskDiT-Pure | WRN-28-10 | **93.11** | **85.41** |

Table 11 presents the comparison of clean and robust accuracy under BPDA attack ($\epsilon = \frac{8}{255}$) on the CIFAR-10 dataset. While the conventional Diffpure method achieves decent robustness, our proposed methods demonstrate significant improvements in both clean and robust accuracy. In particular, RMaskDiT-Pure achieves the highest clean accuracy (93.11%) and robust accuracy (85.41%), highlighting its superior purification capability and enhanced resistance to adversarial attacks. These results validate the effectiveness of our approach in improving semantic reconstruction and adversarial robustness.

## D.9 Ablation study

### D.9.1 Impact of time step number

To investigate the impact of time steps on the denoising process in MaskDiT, we conduct experiments by observing the robust accuracy at different time steps, aiming to understand how varying time steps influence the model's ability to effectively remove noise and improve overall performance.

Table 12: Impact of time step on CIFAR-10, and all configurations align with Table 1.

| Time steps | 15 | 20 | 25 | 30 |
|---|---|---|---|---|
| MaskDiT-Pure | 49.27 | 50.41 | 50.57 | **51.69** |
| RMaskDiT-Pure | 49.22 | 51.34 | **62.13** | 58.22 |

As shown in Table 12, our method exhibits a stable increase in robust accuracy as time steps increase, peaking at **51.69** when the metric of time steps is set as 30. In contrast, RMaskDiT-Pure achieves its highest accuracy of **62.13** when the metric of time steps is set as 25, but experiences a slight drop at the $30^{th}$ step, indicating its sensitivity to the optimal time step selection.

### D.10 Sensitivity Analysis

In this subsection, taking the CIFAR10 under same setting with Table 1, we analyze the impact of step size, mask ratio and step size. The result is described in Fig. 5. It confirms our theoretical analysis.

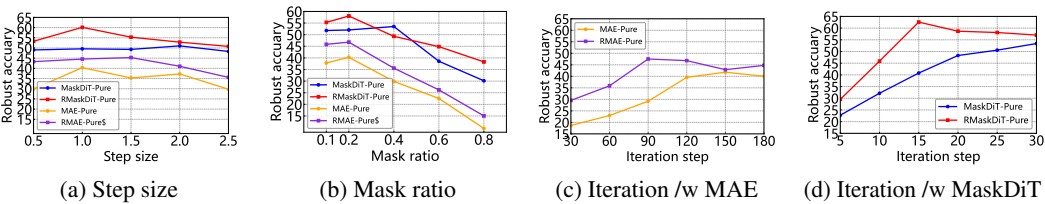

| (a) Step size | (b) Mask ratio | (c) Iteration /w MAE | (d) Iteration /w MaskDiT |
|---|---|---|---|

Figure 5: Sensitivity analysis

## E Supplementary Related work and Preliminary Knowledge

### E.1 Adversarial Training:

Adversarial training (AT) is a technique that enhances the robustness of a neural network by augmenting training samples with additional adversarial examples [13, 19, 38, 47]. Since AT typically involves a high computational cost, some studies [41, 23, 39] have focused on exploring ways to accelerate the training process by a one-step training strategy. In addition, the diffusion model has been employed for extensive data augmentation for adversarial training in many proposals [40, 14, 33], which enlarged the original dataset and enhanced the robust generalization.

### E.2 Preliminary of Masked Autoencoder (MAE)

MAE [15] is briefly introduced within the context of adversarial robustness in this subsection. A clean input sample $x$, drawn from the dataset $X$, is partitioned into $n$ patch vectors of dimension $d$, forming $\bar{x} \in \mathbb{R}^{n \times d}$. The matrix $\bar{x}$ can be randomly divided into $m = (1 - \rho)n$ masked patch vectors and $(n - m)$ visible patch vectors, where $\rho$ is the mask ratio. MAE uses an encoder-decoder architecture. The encoder, $f(\cdot)$, produces $\mathbf{V}^{enc} \in \mathbb{R}^{m \times d_e}$, where $\mathbf{V}^{enc} = f(x_1)$, and $x_1$ is the visible portion of input $x$. Here, $d_e$ is the dimension of each patch feature in $\mathbf{V}^{enc}$. The decoder, $g(\cdot)$, maps $\mathbf{V}^{enc}$ back to pixel space, producing $\mathbf{V}^{dec} \in \mathbb{R}^{(n-m) \times d}$, i.e., $g(\mathbf{V}^{enc}) = \mathbf{V}^{dec}$, which reconstructs masked patches $x_2$. Reconstruction quality is measured using Mean Squared Error (MSE) loss as follows:

$$\mathcal{L}_{\text{rec}}(x_1) = \frac{1}{N(n - m)} \sum_{i=1}^{N} ||g(f(x_{1,i})) - x_{2,i}||^2, \tag{4}$$

where $N$ represents the sample number in $X$.

The MAE structure consists of multiple self-attention layers, where attention captures semantic relationships between input patches. At the $t$-th layer, the input features are $\mathbf{Z}^t \in \mathbb{R}^{n_t \times d_t}$, with $n_t$

patches and $d_t$-dimensional patch features. Weight matrices $\mathbf{W}_Q^t$, $\mathbf{W}_K^t$, and $\mathbf{W}_V^t$ generate the query $\mathbf{Q}^t$, key $\mathbf{K}^t$, and value $\mathbf{V}^t$ matrices, all in $\mathbb{R}^{n_t \times d_t}$.

The self-attention matrix $\mathbf{A}^t$ is computed as:

$$\mathbf{A}^t = \text{softmax} \left( \frac{\mathbf{Q}^t (\mathbf{K}^t)^T}{\sqrt{d_t}} \right),$$

quantifying similarities between $\mathbf{Q}^t$ and $\mathbf{K}^t$. The $j$-th output patch feature $e_j$ is a weighted sum of value vectors:

$$e_j = \sum_{i=1}^{n} a_{ji}^t v_i^t, \quad a_{ji}^t = \frac{q_{ji}^t k_{ji}^t}{\sum_{o=1}^{n} q_{jo}^t},$$

where $a_{ji}^t$ indicates how much $v_i^t$ contributes to $e_j$.

# F  Robust MAE and MaskDiT analysis

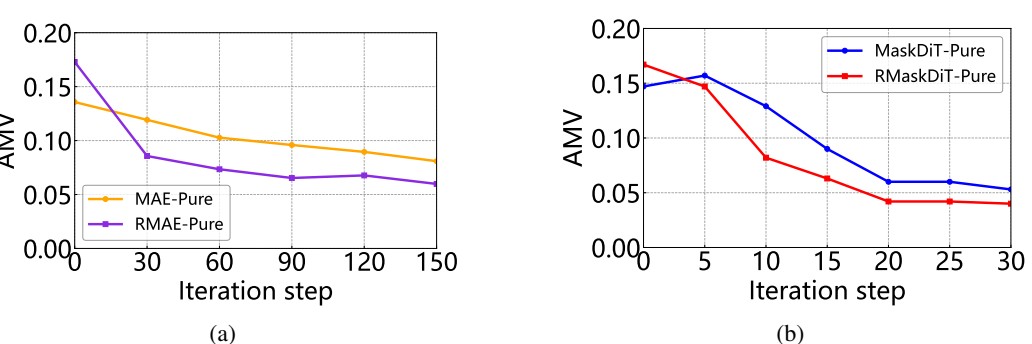

Figure 6: Impact of purification iterations on the AMV using CIFAR-10.

## F.1  Fine-tuning of RMAE-Pure

**Stage I:** We begin by generating adversarial example dataset $\mathrm{X}'_{adv}$, and it achieves the purifier-classifier system as follows:

$$\mathrm{X}'_{adv} = \max_{\delta} \left[ \sum_{(x,y) \in \mathrm{X}} \mathcal{L}_{\mathcal{C}} \left( \mathcal{P}_\theta(x' + \delta), y \right) \right], \tag{5}$$

where $x'$ represents the perturbed sample $x$ with added Gaussian noise, and $y$ denotes its corresponding label from the training dataset X. The functions $\mathcal{P}_\theta(\cdot)$ and $\mathcal{L}_{\mathcal{C}}(\cdot)$ correspond to the purification process and the loss of the classifier, respectively. $x'_{adv}$ is an adversarial sample designed to target the entire purifier-classifier system. It is utilized during the subsequent fine-tuning stage to improve the overall robustness of the system.

**Stage II:** We choose to use the generated adversarial example $x'_{adv}$ from adversarial dataset $\mathrm{X}'_{adv}$ for fine-tuning the purification model as follows:

$$\min_\theta \mathcal{L}_{fine}(x'_{adv}, y, \theta) = \min_\theta \sum_{(x'_{adv}, y) \in \mathrm{X}'_{adv}} \mathcal{L}_{\mathcal{C}} \left( x'_{adv}, y \right), \tag{6}$$

where $\theta$ represents the weight of the purification model.

Compared to MAE-Pure, RMAE-Pure optimizes Eq. (5) to steer denoised images toward the classifier domain of natural datasets. This operation ensures that the attention distribution of denoised images more closely resembles that of natural samples, thereby reducing AMV and preserving inter-patch semantic relationships. As a result, RMAE-Pure achieves superior robustness and generalization.

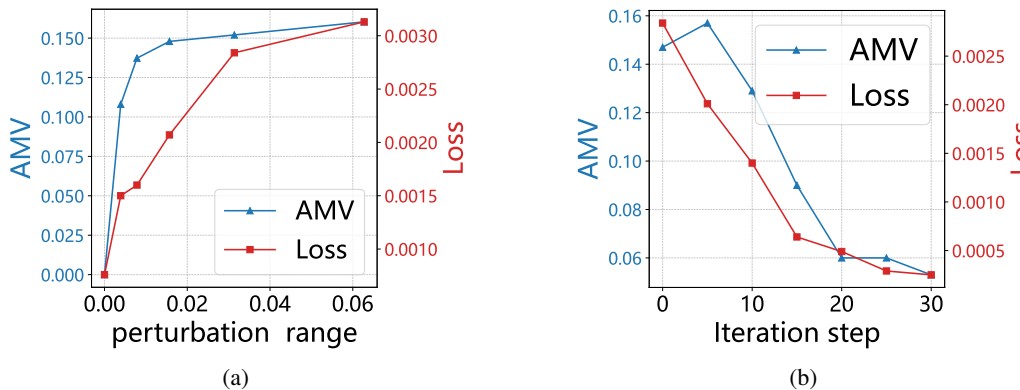

Figure 7: The relationship between the trends of MaskDit loss, AMV, and perturbation.

## G  Validation of MaskDiT

Figure 7 compares the reconstruction loss, AMV, and perturbation of MaskDiT with those of MAE shown in Figures 4, revealing similar patterns. Like MAE, MaskDiT is highly sensitive to noise, with AMV increasing sharply under minimal perturbations (e.g., $\delta = \frac{1}{255}$). Based on this observation, we extend MAE-Pure manner to MaskDiT.

## H  Algorithm

Algorithm 1 describes the MAE-Pure defense procedure. The fine-tuning process of RMAE-Pure consists of two stages.

---
**Algorithm 1** MAE-Pure.

---
**Input:** Adversarial Example $x_{adv}$, Step Size $\lambda$, Number of iteration $S$, clipping threshold $\eta$.
**Output:** Denoised data $x_{den}$.
  1: s ← 0
  2: $x_{adv}^s \leftarrow x_{adv}$
  3: **while** s ≤ S **do**
  4:     s ← s + 1
  5:     Gain the MAE reconstruction loss for adversarial examples $\mathcal{L}_{\text{rec}}(x_{adv}^s)$
  6:     $\Delta_s = \text{sign}(\nabla_x \mathcal{L}_{\text{rec}}(x_{adv}^s))$
  7:     $x_{adv}^s \leftarrow clip(x_{adv}^s - \lambda\Delta_s, \eta)$
  8: **end while**
  9: $x_{den} \leftarrow x_{adv}^S$

---

## I  Visualization Analysis

The visualization analysis of our proposed method is illustrated as Fig. 8. It qualitatively verifies the effectiveness of our proposed method.

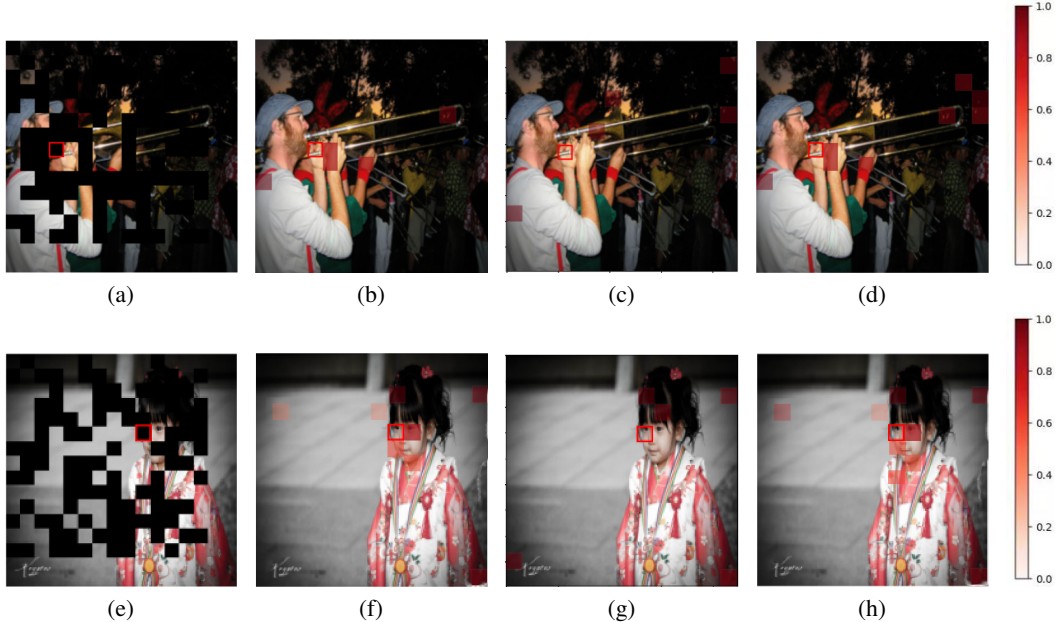

Figure 8: The first column, Fig. (a) and (e), represents the Mask Matrix. The second column, Fig. (b) and (f), illustrates the Attention Weights for clean samples. The third column, Fig. (c) and (k), depicts the Attention Weights for adversarial examples. The fourth column, Fig. (d) and (l), showcases the Attention Weights for denoised samples (by our MAE-Pure). Patches with a deeper red color mean the elements with more attention. The data is sampled from the ImageNet dataset [11].

## J  Convergence Analysis of Purification Process.

For a well-trained MAE model, the reconstruction loss $\mathcal{L}_{\text{rec}}(x)$ is expected to reach a local minimum $\mathcal{L}^*_{\text{rec}}$, where the input $x$ corresponds to a clean example, i.e., $\mathcal{L}_{\text{rec}}(x) \approx \mathcal{L}^*_{\text{rec}}$. Motivated by prior analysis [20, 49] that the loss landscape of MAE is smoother and exhibits wider convex regions, the reconstruction loss $\mathcal{L}_{\text{rec}}$ can be regarded as weakly convex within the neighborhood $[x - \delta, x + \delta]$ around the clean input $x$. Purified samples at the $s$-th iteration of the denoising process are represented as $x^s_{adv}$, and their corresponding reconstruction loss is denoted as $\mathcal{L}^s_{\text{rec}} = \mathcal{L}_{\text{rec}}(x^s_{adv}, \mathbf{U})$, where $\mathbf{U} \in \mathbb{R}^{n \times n}$ represents the MAE mask.

**Theorem J.1.** *Let $\{\mathbf{U}_i\}_{i=1}^E$ be mask set which contains all possible masks with mask ratio $\rho$. After $S$ optimization iterations according to Eq.3, it holds that:*

$$\frac{1}{(1-\rho)E} \sum_{e=1}^E [\mathcal{L}_{rec}(\frac{1}{S+1} \sum_{s=0}^S x^s_{adv}, \mathbf{U}_e) - \mathcal{L}^*_{rec}] \leq \frac{1}{(1-\rho)E} \sum_{e=1}^E [\frac{||x_{adv} - x||_2^2}{2\lambda(S+1)} + \frac{\lambda}{2(S+1)} \sum_{s=0}^S ||\nabla_x \mathcal{L}_{rec}(x^s_{adv}, \mathbf{U}_e)||_2^2],$$

*where $\lambda$ is the step size and $\eta$ is the clipping threshold.*

*Proof.* The proof is relegated to Supplementary Material I.4. □

Theorem J.1 provides an upper bound on the gap between the averaged reconstruction loss during purification and the optimal loss $\mathcal{L}^*_{\text{rec}}$. As $S$ increases, the first term vanishes at $\mathcal{O}(1/S)$ and the second term decreases as the gradient norm $\|\nabla_x \mathcal{L}_{\text{rec}}(x^s_{adv}, \mathbf{U}_e)\|_2^2$ diminishes. Hence, the RHS tends to zero, implying that the LHS also converges to zero. As a result, $\mathcal{L}_{\text{rec}}\left(\frac{1}{S+1} \sum_{s=0}^S x^s_{adv}, \mathbf{U}_e\right) \rightarrow \mathcal{L}^*_{\text{rec}}(x)$, indicating that the denoised sample progressively approximates the clean example in the reconstruction space.

## K  Proof of Theoretical Analysis

### K.1  Assumption

**Assumption 1.** *There exists a pseudo-inverse encoder $f_g$ that satisfies $\|g(f_g(a)) - a\|_2 \le c_{rec}$ for any non-degenerate decoder, where $a$ can be ether $x_1$ or $x_2$.* Here, $x_1$ and $x_2$ indicate visible portion and masked patches of input image $x$. $c_{rec}$ is reconstruction bias, which symbolizes the disparity between the output of MAE and the original, unmasked image.

**Remark.** To facilitate the theoretical analysis of MAE, we borrow the above reasonable assumption from the previous study [49]. Intuitively, this assumption states that within the MAE framework, the decoder $g$ is non-degenerate—i.e., its outputs retain meaningful information—and there exists a corresponding pseudo-inverse encoder $f_g$, such that their composition $h_g = g \circ f_g$ can approximately reconstruct either the visible patches $x_1$ or the masked patches $x_2$ of the input image. Physically, this implies that the decoder in MAE has sufficient capacity for recovering local structures from latent representations, a property empirically verified in many Transformer-based autoencoding models. This assumption provides the theoretical foundation for connecting MAE's reconstruction loss with alignment loss and helps interpret MAE as implicitly performing contrastive alignment through its masking mechanism.

**Assumption 2 (Lipschitz Continuity).** Let $\mathbf{A}_i^{\mathrm{dec}} = \Phi\big(\{\mathbf{A}_{i,t}^{\mathrm{dec}}\}_{t=1}^T\big)$ denote the final decoder attention matrix obtained from the per-layer attention matrices $\{\mathbf{A}_{i,t}^{\mathrm{dec}}\}_{t=1}^T$. We assume $\Phi$ is $L$-Lipschitz in the domain of interest, i.e., there exists a constant $L > 0$ such that for any sets of matrices $\{\mathbf{X}_t\}_{t=1}^T$ and $\{\mathbf{Y}_t\}_{t=1}^T$:

$$\big\|\Phi(\mathbf{X}_1, \ldots, \mathbf{X}_T) - \Phi(\mathbf{Y}_1, \ldots, \mathbf{Y}_T)\big\| \le L\sum_{t=1}^T \|\mathbf{X}_t - \mathbf{Y}_t\|.$$

Under this assumption, for the adversarially perturbed attention matrices $\{\mathbf{A}_{\mathrm{adv},i,t}^{\mathrm{dec}}\}$, the final attention matrix satisfies:

$$\big\|\mathbf{A}_{\mathrm{adv},i}^{\mathrm{dec}} - \mathbf{A}_i^{\mathrm{dec}}\big\|^2 \le L^2 T \sum_{t=1}^T \big\|\mathbf{A}_{\mathrm{adv},i,t}^{\mathrm{dec}} - \mathbf{A}_{i,t}^{\mathrm{dec}}\big\|^2.$$

Hence, we can further write as:

$$\big\|\mathbf{A}_{\mathrm{adv},i}^{\mathrm{dec}} - \mathbf{A}_i^{\mathrm{dec}}\big\|^2 \le \frac{H}{T}\sum_{t=1}^T \big\|\mathbf{A}_{\mathrm{adv},i,t}^{\mathrm{dec}} - \mathbf{A}_{i,t}^{\mathrm{dec}}\big\|^2,$$

by setting $H = L^2 T^2$, thereby bounding the overall adversarial effect via the per-layer differences.

### K.2  Proof of Theorem 3.1

*Proof:*

$$\|\mathbf{A}_{\mathrm{adv}}^t - \mathbf{A}^t\|_2 = \big\|\mathrm{softmax}(\mathbf{Q}_{\mathrm{adv}}^t (\mathbf{K}_{\mathrm{adv}}^t)^T) - \mathrm{softmax}(\mathbf{Q}^t (\mathbf{K}^t)^T)\big\|_2$$

To streamline the equation, we employ kernel methods as a substitute for the softmax$(\cdot)$ function [8]. Let the kernel function be denoted as $\phi(\cdot)$:

$$\mathrm{softmax}(\mathbf{Q}\mathbf{K}^T) \approx \langle \phi(\mathbf{Q}), \phi(\mathbf{K})\rangle$$

The definition is written as follows:

$$\phi(x) = \frac{d(x)}{\sqrt{k}}\{f(\omega_1^\top x), ..., f(\omega_k^\top x)\}, \quad d(x) = \exp\left(-\frac{\|x\|^2}{2}\right), \quad f(x) = \exp(x)$$

Thus, we obtain:

$$\mathbf{A}_{\text{adv}}^t - \mathbf{A}^t \approx \frac{1}{m}\left[\exp\left(-\frac{||\mathbf{Q}_{\text{adv}}^t||^2 + ||\mathbf{K}_{\text{adv}}^t||^2}{2}\right)\sum_{i=0}^{k}\exp\left(\omega_i^\top(\mathbf{Q}_{\text{adv}}^t + \mathbf{K}_{\text{adv}}^t)\right)\right.$$
$$\left. - \exp\left(-\frac{||\mathbf{Q}^t||^2 + ||\mathbf{K}^t||^2}{2}\right)\sum_{i=0}^{k}\exp\left(\omega_i^\top(\mathbf{Q}^t + \mathbf{K}^t)\right)\right]$$

Assume: $\mathbf{Q}_{\text{adv}}^t = \mathbf{Q}^t + \Delta\mathbf{Q}^t$, $\mathbf{K}_{\text{adv}}^t = \mathbf{K}^t + \Delta\mathbf{K}^t$

Using first-order approximations, the formulation can be deduced as follows:

$$\exp\left(-\frac{||\mathbf{Q}_{\text{adv}}^t||^2 + ||\mathbf{K}_{\text{adv}}^t||^2}{2}\right) \approx \exp\left(-\frac{||\mathbf{Q}^t||^2 + ||\mathbf{K}^t||^2}{2}\right)\left(1 - (\mathbf{Q}^t)^\top\Delta\mathbf{Q}^t - (\mathbf{K}^t)^\top\Delta\mathbf{K}^t\right)$$

Let $\mathbf{S} = \mathbf{Q}^t + \mathbf{K}^t$, $\Delta\mathbf{S} = \Delta\mathbf{Q}^t + \Delta\mathbf{K}^t$, there exists a following equation:

$$\sum_{i=0}^{k}\exp\left(\omega_i^\top(\mathbf{S} + \Delta\mathbf{S})\right) \approx \sum_{i=0}^{k}\exp\left(\omega_i^\top\mathbf{S}\right) + \sum_{i=0}^{k}\exp\left(\omega_i^\top\mathbf{S}\right)\omega_i^\top\Delta\mathbf{S}$$

The definition is written as follows:

$$\mathbf{B} = \sum_{i=0}^{k}\exp\left(\omega_i^\top\mathbf{S}\right), \quad \mathbf{Y} = \sum_{i=0}^{k}\exp\left(\omega_i^\top\mathbf{S}\right)\omega_i$$

Then, we deduce the following sum equation:

$$\sum_{i=0}^{k}\exp\left(\omega_i^\top(\mathbf{Q}_{\text{adv}}^t + \mathbf{K}_{\text{adv}}^t)\right) \approx \mathbf{B} + \mathbf{Y}^\top\Delta\mathbf{S} = \mathbf{B} + \mathbf{Y}^\top(\Delta\mathbf{Q}^t + \Delta\mathbf{K}^t)$$

The substitution into difference is:

$$D \approx \frac{1}{m}\exp\left(-\frac{||\mathbf{Q}^t||^2 + ||\mathbf{K}^t||^2}{2}\right)\left[\left(1 - (\mathbf{Q}^t)^\top\Delta\mathbf{Q}^t - (\mathbf{K}^t)^\top\Delta\mathbf{K}^t\right)\left(\mathbf{B} + \mathbf{Y}^\top(\Delta\mathbf{Q}^t + \Delta\mathbf{K}^t)\right) - \mathbf{B}\right]$$
$$\approx \frac{1}{m}\exp\left(-\frac{||\mathbf{Q}^t||^2 + ||\mathbf{K}^t||^2}{2}\right)\left[\mathbf{Y}^\top(\Delta\mathbf{Q}^t + \Delta\mathbf{K}^t) - \mathbf{B}\left((\mathbf{Q}^t)^\top\Delta\mathbf{Q}^t + (\mathbf{K}^t)^\top\Delta\mathbf{K}^t\right)\right]$$

We group and finalize to get a concluding formulation:

$$||\mathbf{A}_{\text{adv}}^t - \mathbf{A}^t||_2 \approx \frac{\exp\left(-\frac{||\mathbf{Q}^t||^2 + ||\mathbf{K}^t||^2}{2}\right)}{m}\left\|\left(\mathbf{Y} - \mathbf{B}\mathbf{Q}^t\right)^\top\Delta\mathbf{Q}^t + \left(\mathbf{Y} - \mathbf{B}\mathbf{K}^t\right)^\top\Delta\mathbf{K}^t\right\|_2$$
$$= \gamma\left\|\left(\mathbf{Y} - \mathbf{B}\mathbf{Q}^t\right)^\top\Delta\mathbf{Q}^t + \left(\mathbf{Y} - \mathbf{B}\mathbf{K}^t\right)^\top\Delta\mathbf{K}^t\right\|_2$$

where the variables' definitions are:

- $\mathbf{B} = \sum_{i=0}^{k}\exp\left(\omega_i^\top(\mathbf{Q}^t + \mathbf{K}^t)\right)$,
- $\mathbf{Y} = \sum_{i=0}^{k}\exp\left(\omega_i^\top(\mathbf{Q}^t + \mathbf{K}^t)\right)\omega_i$,
- $\gamma = \frac{\exp\left(-\frac{||\mathbf{Q}^t||^2 + ||\mathbf{K}^t||^2}{2}\right)}{m}$.

The proof is complete.

## K.3 Proof of Theorem 3.2

*Proof:*

$$\mathcal{L}_{\text{rec}}^{adv} = \frac{1}{N}\sum_{i=1}^{N}||g(f(x_{i_1}^{adv})) - x_{i_2}^{adv}||^2$$

$$= \frac{1}{N}\sum_{i=1}^{N}[||g(f(x_{i_1}^{adv})) - x_{i_2}^{adv}||_2 + c_{\text{rec}} - c_{\text{rec}}]$$

$$= \frac{1}{N}\sum_{i=1}^{N}[||g(f(x_{i_1}^{adv})) - x_{i_2}^{adv}||^2 + ||g(f_g(x_{i_2}^{adv})) - x_{i_2}^{adv}||^2 - c_{\text{rec}}]$$

$$= \frac{1}{N}\sum_{1}^{N}[||g(f(x_{i_1}^{adv})) - x_{i_2}^{adv} + g(f(x_{i_1})) - g(f(x_{i_1}))||^2 + ||g(f_g(x_2)) - x_2||^2 - c_{\text{rec}}]$$

$$\geq \frac{1}{N}\sum_{i=1}^{N}[\frac{1}{2}||g(f(x_{i_1}^{adv})) - x_{i_2}^{adv} + g(f(x_{i_1})) - g(f(x_{i_1})) + g(f_g(x_2)) - x_2||^2 - c_{\text{rec}}]$$

$$\geq \frac{1}{2N}\sum_{i=1}^{N}[||g(f(x_{i_1}^{adv})) - x_{i_2}^{adv} + g(f(x_{i_1})) - g(f(x_{i_1})) + g(f_g(x_2)) - x_2||^2 - 2c_{\text{rec}}]$$

$$\geq \frac{1}{2N}\sum_{i=1}^{N}[||g(f(x_{i_1})) - x_{i_2}||^2 + ||g(f(x_{i_1}^{adv})) - g(f(x_{i_1}))||^2 + ||g(f_g(x_{i_2}) - x_{i_2}^{adv}||^2 - 2c_{\text{rec}}]$$

$$\geq \frac{1}{2}\mathcal{L}_{\text{rec}} + \frac{1}{2N}\sum_{i=1}^{N}[||g(f(x_{i_1}^{adv})) - g(f(x_{i_1}))||^2 + ||g(f_g(x_{i_2}) - x_{i_2}^{adv}||^2 - 2c_{\text{rec}}].$$

Since we have $||g(f_g(x_{i_2}) - x_{i_2}^{adv}||^2 = ||g(f_g(x_{i_2}) - x_{i_2} + \delta||^2$, and $||\delta||_2$ is tiny, it leads $||g(f_g(x_{i_2}) - x_{i_2} + \delta||^2 \geq ||g(f_g(x_{i_2}) - x_{i_2})||^2 - ||\delta||^2$. Following the previous study [4], the decoder $g(\cdot)$ in a MAE can be represented as an interpolation of the encoder output, denoted by $g(f(x)) = \mathbf{A}_{\text{dec}}\mathbf{V}_{\text{enco}}$. $A_{\text{dec}}$ represents the interpolation weights, and $\mathbf{V}_{\text{enco}}$ is the encoder output.

$$\mathcal{L}_{\text{rec}}^{adv} \geq \frac{1}{2}\mathcal{L}_{\text{rec}} + \frac{1}{2N}\sum_{i=1}^{N}[||\mathbf{A}_{adv_i}^{dec}\mathbf{V}_{adv_i}^{enco} - \mathbf{A}_i^{dec}\mathbf{V}_i^{enco}||^2 - ||\delta||^2 - c_{\text{rec}}]$$

$$= \frac{1}{2}\mathcal{L}_{\text{rec}} + \frac{1}{2N}\sum_{i=1}^{N}[||\mathbf{A}_{adv_i}^{dec}\mathbf{V}_{adv_i}^{enco} - \mathbf{A}_i^{dec}\mathbf{V}_i^{enco} + \mathbf{A}_i^{dec}\mathbf{V}_{adv_i}^{enco} - \mathbf{A}_i^{dec}\mathbf{V}_{adv_i}^{enco}||^2 - ||\delta||^2 - c_{\text{rec}}]$$

$$\geq \frac{1}{2}\mathcal{L}_{\text{rec}} + \frac{1}{2N}\sum_{i=1}^{N}[||\mathbf{V}_{adv_i}^{enco}(\mathbf{A}_{adv_i}^{dec} - \mathbf{A}_i^{dec})||^2 + ||\mathbf{A}_i^{dec}(\mathbf{V}_i^{enco} - \mathbf{V}_{adv_i}^{enco})||^2 - ||\delta||^2 - c_{\text{rec}}].$$

Since $||\mathbf{A}_i^{dec}(\mathbf{V}_i^{enco} - \mathbf{V}_{adv_i}^{enco})||^2 \geq ||\delta||^2$ and a ratio constant $C_A$, we can get the following equation:

$$\mathcal{L}_{\text{rec}}^{adv} \geq \frac{1}{2}\mathcal{L}_{\text{rec}} + \frac{1}{2N}\sum_{i=1}^{N}[C_A||(\mathbf{A}_{adv_i}^{dec} - \mathbf{A}_i^{dec})||^2 - c_{\text{rec}}].$$

Based on Assumption 2, there exists a constant $H$ such that $\left\|\mathbf{A}_{\text{adv},i}^{\text{dec}} - \mathbf{A}_i^{\text{dec}}\right\|^2 \approx \frac{H}{T}\sum_{t=1}^{T}\left\|\mathbf{A}_{\text{adv},i,t}^{\text{dec}} - \mathbf{A}_{i,t}^{\text{dec}}\right\|^2$. $T$ represents the number of layers in the decoder, and $\mathbf{A}_{adv_i,t}^{dec}$ and $\mathbf{A}_{i,t}^{dec}$ are the attention matrices at the $t$-th decoder layer corresponding to $i$-th adversarial example and the clean example, respectively. Therefore, we can conclude the proof:

$$\mathcal{L}_{\text{rec}}^{adv} \geq \frac{1}{2}\mathcal{L}_{\text{rec}} + \frac{1}{2NT}\sum_{t=1}^{T}\sum_{i=1}^{N}[HC_A||(\mathbf{A}_{adv_i,t}^{dec} - \mathbf{A}_{i,t}^{dec})||^2 - c_{\text{rec}}].$$

764  The proof is complete.

### K.4  Proof of Theorem 4.1

766  *Proof:*

767  We assume $\mathcal{L}_{\text{rec}}$ function is convex at the area $[x - \delta, x + \delta]$. For fixed mask $U_0$, we can get the
768  following inequality:

$$\mathcal{L}_{\text{rec}}(x_{adv}, U_0) \leq \mathcal{L}_{\text{rec}}(x, U_0) + < \nabla_x L(x, U_0), x_{adv} - x >$$
$$\iff \mathcal{L}_{\text{rec}}(x_{adv}, U_0) - L_{\text{rec}}(x, U_0) \leq < \nabla_x L(x, U_0), x_{adv} - x > .$$

We define $y_{adv}^1 = x_{adv} - \nabla_x \mathcal{L}(x_{adv}, U_0)$,

$$\iff \mathcal{L}_{\text{rec}}(x_{adv}, U_0) - \mathcal{L}_{\text{rec}}(x, U_0) \leq < \frac{x_{adv} - y_{adv}^1}{\lambda}, x_{adv} - x >$$

$$\Rightarrow \mathcal{L}_{\text{rec}}(x_{adv}, U_0) - \mathcal{L}_{\text{rec}}(x, U_0) \leq \frac{< x_{adv} - y_{adv}^1, x_{adv} - x >}{\lambda}$$

$$= \frac{(x_{adv})^2 - x_{adv}x - x_{adv}y_{adv}^1 + y_{adv}^1 x}{\lambda}$$

$$= \frac{2(x_{adv})^2 - 2x_{adv}x - 2x_{adv}y_{adv}^1 + 2y_{adv}^1 x}{2\lambda}$$

$$\frac{(X_{adv})^2 - 2x_{adv}x + (x_{adv})^2 - 2y_{adv}x_{adv}^1 + x^2 - x^2 + (y_{adv}^1)^2 - (y_{adv}^1)^2}{2\lambda}$$

$$= \frac{||x_{adv} - x||_2^2 + ||x_{adv} - y_{adv}^1||_2^2 - ||y_{adv}^1 - x||_2^2}{2\lambda}.$$

769  Therefore, we can get the inequality:

$$\mathcal{L}_{\text{rec}}(x_{adv}, U_0) - \mathcal{L}_{\text{rec}}(x, U_0) \leq \frac{||x_{adv} - x||_2^2 - ||y_{adv}^1 - x||_2^2}{2\lambda} + \frac{\lambda}{2}||\nabla_X \mathcal{L}(x_{adv}, U_0)||_2^2.$$

770  Since $||y_{adv}^1 - x||_2^2 \leq ||clip(y_{adv}^1, \eta) - x||_2^2$, and $\eta$ is the clipping threshold, we define $x_{adv}^1 =$
771  $clip(y_{adv}^1, \eta)$ and get a derivation:

$$\mathcal{L}_{\text{rec}}(x_{adv}, U_0) - \mathcal{L}_{\text{rec}}(x, U_0) \leq \frac{||x_{adv} - x||_2^2 - ||x_{adv}^1 - x||_2^2}{2\lambda} + \frac{\lambda}{2}||\nabla_x \mathcal{L}(x_{adv}, U_0)||_2^2.$$

772  As the same theory, the law is written as the following formulation:

$$\begin{cases} s = 0 \\ \mathcal{L}_{\text{rec}}(x_{adv}, U_0) - \mathcal{L}_{\text{rec}}(x, U_0) \leq \frac{||x_{adv} - x||_2^2 - ||x_{adv}^1 - x||_2^2}{2\lambda} + \frac{\lambda}{2}||\nabla_X \mathcal{L}(x_{adv}^1, U_0)||_2^2 \\ s = 1 \\ \mathcal{L}_{\text{rec}}(x_{adv}^1, U_0) - \mathcal{L}_{\text{rec}}(x, U_0) \leq \frac{||x_{adv} - x||_2^2 - ||x_{adv}^2 - x||_2^2}{2\lambda} + \frac{\lambda}{2}||\nabla_x \mathcal{L}(x_{adv}^2, U_0)||_2^2 \\ s = 2 \\ \mathcal{L}_{\text{rec}}(x_{adv}^2, U_0) - \mathcal{L}_{\text{rec}}(x, U_0) \leq \frac{||x_{adv} - x||_2^2 - ||x_{adv}^3 - x||_2^2}{2\lambda} + \frac{\lambda}{2}||\nabla_x \mathcal{L}(x_{adv}^3, U_0)||_2^2 \\ \qquad\qquad\qquad ...... \\ s = S \\ \mathcal{L}_{\text{rec}}(x_{adv}^S, U_0) - \mathcal{L}_{\text{rec}}(x, U_0) \leq \frac{||x_{adv} - x||_2^2 - ||x_{adv}^{S+1} - x||_2^2}{2\lambda} + \frac{\lambda}{2}||\nabla_x \mathcal{L}(x_{adv}^S, U_0)||_2^2 \end{cases}$$

773  We sum all the above terms as follows:

$$\sum_{s=0}^{S}(\mathcal{L}_{\text{rec}}(x_{adv}^s, U_0) - \mathcal{L}_{\text{rec}}(x, U_0)) \leq \frac{||x_{adv} - x||_2^2 - ||x_{adv}^{S+1} - x||_2^2}{2\lambda} + \frac{\lambda}{2}\sum_{s=0}^{S}||\nabla_x \mathcal{L}(x_{adv}^s, U_0)||_2^2$$

$$\leq \frac{||x_{adv} - x||_2^2}{2\lambda} + \frac{\lambda}{2}\sum_{s=0}^{S}||\nabla_x \mathcal{L}(x_{adv}^s, U_0)||_2^2$$

The average loss upper bound $\frac{1}{S+1}\sum_{s=0}^{S}(\mathcal{L}_{\text{rec}}(x_{adv}^s, U_0) - \mathcal{L}_{\text{rec}}(x, U_0))$ is given as the following inequality:

$$\frac{1}{S+1}\sum_{s=0}^{S}(\mathcal{L}_{\text{rec}}(x_{adv}^s, U_0) - \mathcal{L}_{\text{rec}}(x, U_0)) \leq \frac{||x_{adv} - x||_2^2}{2\lambda(S+1)} + \frac{\lambda}{2(S+1)}\sum_{s=0}^{S}||\nabla_x\mathcal{L}(x_{adv}^s, U_0)||_2^2.$$

Due to the weak convexity of $\mathcal{L}_{\text{rec}}(\cdot)$, it is evident that $\frac{1}{S+1}\sum_{s=0}^{S}\mathcal{L}_{\text{rec}}(x_{\text{adv}}^s, U_0) \geq \mathcal{L}_{\text{rec}}\left(\frac{1}{S+1}\sum_{s=0}^{S}X_{\text{adv}}^s, U_0\right)$ and we can obtain the formulation as follows:

$$\mathcal{L}_{\text{rec}}\left(\frac{1}{S+1}\sum_{s=0}^{S}x_{\text{adv}}^s, U_0\right) - \mathcal{L}_{\text{rec}}(x, U_0)) \leq \frac{||x_{adv} - x||_2^2}{2\lambda(S+1)} + \frac{\lambda}{2(S+1)}\sum_{s=0}^{S+1}||\nabla_x\mathcal{L}(x_{adv}^s, U_0)||_2^2.$$

For any mask $U_e$ and $e \in [1, E]$, we can obtain similar results. Thus, we can draw the results:

$$\frac{1}{(1-\rho)E}\sum_{e=1}^{E}[\mathcal{L}_{\text{rec}}\left(\frac{1}{S+1}\sum_{s=0}^{S}x_{\text{adv}}^s, U_e\right) - \mathcal{L}_{MAE}(x, U_e))] \leq \frac{1}{(1-\rho)E}\sum_{e=1}^{E}[\frac{||x_{adv} - x||_2^2}{2\lambda(S+1)}$$
$$+ \frac{\lambda}{2(S+1)}\sum_{s=0}^{S}||\nabla_x\mathcal{L}(x_{adv}^s, U_e)||_2^2].$$

For a fixed masked rate $p$, the proof has been completed. $\mathcal{L}_{\text{rec}}(X, U_e)) \approx \mathcal{L}_{\text{rec}}^*(X)$ for any $e \in [1, E]$, and we can conclude the proof:

$$\frac{1}{(1-\rho)E}\sum_{e=1}^{E}[\mathcal{L}_{\text{rec}}\left(\frac{1}{S+1}\sum_{s=0}^{S}x_{\text{adv}}^s, U_e\right) - \mathcal{L}_{\text{rec}}^*(x)] \leq \frac{1}{(1-\rho)E}\sum_{e=1}^{E}[\frac{||x_{adv} - x||_2^2}{2\lambda(S+1)}$$
$$+ \frac{\lambda}{2(S+1)}\sum_{s=0}^{S+1}||\nabla_x\mathcal{L}(x_{adv}^s, U_e)||_2^2].$$

The proof is complete.

# L  Details of Mask Autoencoder

**MAE:** For CIFAR-10, CIFAR-100, and SVHN, we use Base-MAE [15], setting the image size to 32 and the patch size to 4, while leaving the other parameters unchanged. For ImageNet, we directly used Large-MAE.

For MAE training on CIFAR-10, CIFAR-100, and ImageNet, we use A100 GPUs, training for 2000 epochs with the learning rate of $1e^{-3}$, followed by an additional 1000 epochs with the learning rate of $1.5e^{-4}$. Our training batch size is 64. For ImageNet, we directly use the checkpoint provided by the author.

**MaskDiT:** For SVHN, CIFAR100, and CIFAR10, the network first divides the 32x32 CIFAR-10 image into non-overlapping 8x8 patches, with each patch sized 4x4, resulting in a total of 64 patches. Each patch is mapped to a 128-dimensional embedding space through a linear projection layer, and learnable positional encodings are added. During training, 50% of the patches are randomly masked, and only the unmasked patches are fed into an encoder composed of 6 Transformer blocks, each consisting of multi-head self-attention and a feed-forward network. Subsequently, the encoded unmasked patches are concatenated with learnable mask tokens and passed into a decoder composed of 3 lightweight Transformer blocks. Finally, a linear projection layer maps the decoded patches back to the 4x4x3 patch space, completing image reconstruction and denoising score prediction. For ImageNet, we use patch size as $16 \times 16$ and other parameter are same with original work [50]. Both network training process and parameters are aligned.

## L.1  Details of Robust fine-tuning

For CIFAR-10, CIFAR-100, and SVHN, MAE-Pure is fine-tuned for 100 epochs, while for ImageNet, it is fine-tuned for 3 epochs. In contrast, MaskDiT is fine-tuned for 25 epochs on CIFAR-10, CIFAR-100, and SVHN, and 1 epoch on ImageNet.

## L.2 Details of Experiment

### L.2.1 Hyperparameters in Denoising Process

All hyperparameters related to the denoising process are encompassed in Table 13 and 14, covering the number of denoising iterations, step size, mask rate, and patch size for each dataset. The asterisk (*) indicates that the step size decays by 0.5 after more than half of the iterations have been completed.

Table 13: Hyperparameters of Step Size and Patch Size

| Dataset | Step Size | | | | Patch Size | | | |
|---|---|---|---|---|---|---|---|---|
| | MAE | RMAE | MaskDiT | RMaskDiT | MAE | RMAE | MaskDiT | RMaskDiT |
| CIFAR-10 | 1* | 1* | 1 | 1 | 4 | 4 | 4 | 4 |
| CIFAR-100 | 1 | 1 | 1 | 1 | 4 | 4 | 4 | 4 |
| SVHN | 1* | 1 | 1 | 1 | 4 | 4 | 4 | 4 |
| ImageNet | 1* | 1 | 1 | 1 | 16 | 16 | 16 | 16 |

Table 14: Hyperparameters of Mask Rate and Iteration Numbers

| Dataset | Mask Rate | | | | Iterations | | | |
|---|---|---|---|---|---|---|---|---|
| | MAE | RMAE | MaskDiT | RMaskDiT | MAE | RMAE | MaskDiT | RMaskDiT |
| CIFAR-10 | 0.25 | 0.25 | 0.50 | 0.50 | 150 | 90 | 25 | 15 |
| CIFAR-100 | 0.30 | 0.25 | 0.50 | 0.50 | 100 | 65 | 20 | 20 |
| SVHN | 0.30 | 0.25 | 0.50 | 0.50 | 150 | 80 | 20 | 20 |
| ImageNet | 0.20 | 0.25 | 0.30 | 0.30 | 150 | 20 | 25 | 20 |

