# OpenReview forum: "MAE-Pure: Semantic-Preserving Adversarial Purification"
_NeurIPS.cc/2025/Conference — Submitted to NeurIPS 2025_

### Official Review · Reviewer_WB7Q · 2025-06-20

**Clarity:** 4
**Significance:** 3
**Originality:** 4
**Rating:** 4
**Confidence:** 4

**Summary:**

This paper introduces MAE-Pure, a novel adversarial purification method that leverages Masked Autoencoders (MAE) to preserve semantic relationships among image patches for effective defense against adversarial attacks. The key insight is that adversarial perturbations significantly alter the attention matrix of MAE, thereby degrading its reconstruction ability. MAE-Pure formulates purification as an optimization problem minimizing this Attention Matrix Variation (AMV). To extend performance, the authors also adapt the method to diffusion-based models (MaskDiT-Pure) and introduce robust variants (RMAE-Pure, RMaskDiT-Pure) by incorporating classification loss. Theoretical analysis and empirical experiments across CIFAR-10/100, SVHN, and ImageNet show strong robustness and clean accuracy, setting new SOTA results in many benchmarks.

**Questions:**

Can MAE-Pure be adapted to architectures that do not employ attention mechanisms (e.g., CNN autoencoders)? Or is the attention matrix variation uniquely leveraged here?

RMaskDiT-Pure improves robustness significantly, but in some cases (e.g., CIFAR-100), slightly drops clean accuracy. Could the authors comment on how this tradeoff is managed during fine-tuning?

Have you considered whether this AMV-based purification could generalize to non-image modalities (e.g., audio, text) where masking and reconstruction are common?

Would the authors consider releasing tools for visualizing AMV evolution during attacks or purification? This could be useful for broader community analysis.

**Ethical Concerns:**

["NO or VERY MINOR ethics concerns only"]

**Final Justification:**

I have read the author response carefully and check everything, and I would like to keep my score.

**Limitations:**

Yes. The paper openly discusses limitations such as memory consumption with batch size and acknowledges the method's reliance on MAE-like architectures. However, a more in-depth societal reflection could be included.

**Quality:**

3

**Strengths And Weaknesses:**

The paper presents several notable strengths. First, it offers a novel perspective on adversarial purification by focusing on preserving inter-patch semantic consistency through attention matrix analysis—a direction largely unexplored in prior work. The theoretical contributions are strong, with two theorems rigorously establishing how adversarial perturbations influence MAE attention matrices and lead to degraded reconstruction performance. Methodologically, the paper is clear in deriving its optimization framework and justifying the use of reconstruction loss as a tractable surrogate for attention matrix variation minimization. Another strength lies in its extensibility; the authors successfully generalize their approach to diffusion-based models (MaskDiT) and enhance robustness through fine-tuned variants (RMAE-Pure and RMaskDiT-Pure). The empirical validation is thorough, demonstrating consistent improvements across diverse datasets (CIFAR-10/100, SVHN, ImageNet) and attack types, including ℓ∞, ℓ2, ℓ1, BPDA, EOT, and black-box attacks. Moreover, the method is efficient, as it achieves strong robustness without requiring extra data and demonstrates favorable inference time, particularly on large-scale datasets like ImageNet.

However, the paper has some weaknesses. The approach is tightly coupled with the MAE architecture, and it remains unclear whether the insights or methods would generalize to non-attention-based architectures such as standard CNN autoencoders. While the theoretical analysis is sound, the role of reconstruction bias constants (e.g., crec) and the approximation of kernel coefficients (e.g., ω in Theorem 3.1) are not empirically explored in depth, leaving questions about the sensitivity of the theoretical bounds in practice. Additionally, the notation—especially in the theoretical sections—can be dense, with key definitions (e.g., B and Y in Theorem 3.1) placed in the appendix rather than clarified in the main body, which could hinder readability. Lastly, although the broader impact section highlights the novelty of their perspective, it does not sufficiently address potential negative societal implications, such as the use or misuse of such defense techniques in sensitive domains like surveillance or biometric authentication.

---

> ### Author Rebuttal · Authors · 2025-07-31
>
> We sincerely thank the reviewer for the kind and constructive comments, which have greatly helped us improve the quality of our paper. We look forward to further engaging with you during the discussion phase.
>
>
> **W1: Extension experiment on CNN structure**
>
> We selected a CNN architecture similar to MAE, called Context Encoders [1], and proposed CE-Pure in a manner similar to MAE-Pure, to test on CIFAR-10 and SVHN, using WRN-28 as the classifier. Context Encoders are a CNN-based self-supervised learning method that learns contextual features of images by masking regions and reconstructing the missing content. Context Encoders and Masked Autoencoders (MAE) are both self-supervised learning methods based on masking. Their core idea is to mask part of the input and train the model to reconstruct the missing content, thereby learning meaningful feature representations. The main difference between the two lies in their architectural design: Context Encoders are based on CNNs, which are better at capturing local structural information and are commonly used for image inpainting tasks; in contrast, MAE is based on the Transformer architecture, which excels at modeling global contextual information. Context Encoders are a CNN-based self-supervised learning method that learns contextual features of images by masking regions and reconstructing the missing content.
> Alignment with main paper, we use PGD200+EOT20 with full gradient. The results are as follows:
>
> **Table: Comparison of CE-Pure and MAE-Pure results on CIFAR-10 and SVHN using WRN-28 as the classifier.**
>
> | Method   | Std Acc (CIFAR10) | Robust Acc (CIFAR10, $\ell_{\infty}$) | Robust Acc (CIFAR10, $\ell_{2}$) | Std Acc (SVHN) | Robust Acc (SVHN, $\ell_{\infty}$) | Robust Acc (SVHN, $\ell_{2}$) |
> |----------|-------------------|----------------------------------------|----------------------------------|----------------|-------------------------------------|-------------------------------|
> | CE-Pure  | 82.59             | 6.75                                   | 11.59                            | 85.33          | 8.99                                | 14.21                         |
> | MAE-Pure | 88.57             | 40.53                                  | 53.50                            | 95.39          | 55.90                               | 70.18                         |
>
> The results show that the robustness of MAE-Pure is significantly stronger than that of CE-Pure. This mechanism cannot be transferred to CNNs, thus proving that our theory is correct.
>
> **W2: Concerns about kernel coefficients**
>
> Thank you for your valuable feedback. Regarding the reconstruction bias constants (e.g., $C\_\text{rec}$) and the approximation of kernel coefficients (e.g., $ \omega $ in Theorem 3.1), we agree that a deeper empirical exploration of these factors' sensitivity to the theoretical bounds is currently lacking. In future work, we plan to conduct more detailed empirical analysis to better understand the impact of these parameters on the theoretical results and optimize our estimation methods.
>
> Additionally, concerning the notation and the placement of key definitions (e.g., B  and Y  in Theorem 3.1), we understand that this may affect the readability of the paper. In the revision, we will clarify the notation and ensure that key content is not placed in the appendix but is clearly explained in the main body of the text to improve understanding of the theoretical sections.
>
> Once again, thank you for your suggestions. We will make sure to incorporate your feedback and further refine the relevant sections in the revision.
>
> **W3: Concerns about border impact**
>
> We thank the reviewer for highlighting the potential for misuse of our defense techniques in sensitive domains such as surveillance or biometric authentication. To address this concern concretely, we propose the following mitigation strategies:
>
> 1. **Domain-Specific Safeguards:** Our method (e.g., MAE-Pure, RMaskDiT-Pure) is not intended for direct use in surveillance or biometric applications without substantial adaptation. We recommend enforcing access control and fine-tuning with fairness-aware objectives when applied to such domains.
>
> 2. **Auditability and Interpretability:** We will release our codebase with tools for visualizing semantic attention maps before and after purification. This transparency enables third-party auditing and fosters responsible adoption.
>
> 3. **Use-Case Licensing:** We plan to adopt a license (e.g., OpenRAIL-style) that explicitly prohibits use in biometric identification, mass surveillance, or military applications.
>
> 4. **Fairness and Bias Audits:** While our semantic-preserving approach may improve robustness uniformly, we emphasize the need for demographic fairness audits before deployment in human-centric tasks to avoid unintended bias.
>
> 5. **Collaborative Deployment Oversight:** We advocate for collaborative oversight involving ethicists, legal experts, and application-domain practitioners when deploying purification models in safety-critical environments.
>
> These strategies aim to ensure our method supports robustness research while minimizing the risk of unethical deployment. We also will add them in the revision late.
>
> **Q1: Extension to CNN-based autoencoders**
>
> Thank you for the question. As shown in our additional experiments addressing Weakness 1, MAE-Pure’s effectiveness is critically based on attention mechanisms. CNN-based autoencoders as context autoencoders, which lack attention matrices, show very week-round robustness. As a result, the purification does not generalize to CNN architectures. This confirms that attention is not optional, but essential for the design and success of our method.
>
> **Q2: manage trade-off**
> We thank the reviewer for highlighting the slight drop in clean accuracy observed on CIFAR-100 when using RMaskDiT-Pure. This phenomenon stems from the trade-off introduced during the fine-tuning phase, where we incorporate a classification loss to enhance robustness. While this additional supervision effectively guides the model to focus on features invariant under adversarial perturbations, it may unintentionally suppress fine-grained features that are important for clean classification—especially on complex datasets like CIFAR-100, where inter-class distinctions are subtle.
>
> To better manage this trade-off, we are exploring two main strategies. First, we aim to refine the hyperparameters involved in fine-tuning—such as adjusting the weight of the classification loss or employing early stopping based on clean validation accuracy—to prevent over-regularization. Second, we are investigating selective data filtering techniques that can reduce the influence of adversarially ambiguous or noisy samples. This can help the model retain more semantically aligned features during fine-tuning, thereby preserving clean accuracy while maintaining robustness.
>
> **Q3: Extension to other modalities**
>
> We appreciate the reviewer’s thoughtful question regarding the potential generalization of our AMV-based purification framework to non-image modalities such as audio and text.
>
> While our current work is focused on the vision domain, we agree that the core idea of leveraging attention matrix variation (AMV) for detecting and correcting semantic distortions could be applicable to other modalities—especially those that rely on masked reconstruction mechanisms, such as masked language models in NLP (e.g., BERT) or spectrogram-based masked modeling in speech processing. These domains also involve attention-based architectures where inter-token or inter-frame semantic consistency plays a crucial role.
>
> Exploring this cross-modal generalization is a promising direction, and we plan to investigate the extension of our approach to textual and audio data in future work. We believe such studies could further reveal the universality and flexibility of AMV-based semantic purification.
>
> **Q4: Releasing of visualizing tools**
>
> We thank the reviewer for the valuable suggestion. We fully agree that tools for visualizing the evolution of Attention Matrix Variation (AMV) during attacks and purification can be highly beneficial for the research community.
>
> In our final release, we plan to open-source the complete codebase, including all components related to AMV computation and visualization. This will enable researchers to easily reproduce our results, analyze AMV dynamics under various settings, and potentially extend the visualization tools to other tasks and modalities.
>
> [1] Pathak, D., Krahenbuhl, P., Donahue, J., Darrell, T., & Efros, A. A. (2016). Context encoders: Feature learning by inpainting. In Proceedings of the IEEE conference on computer vision and pattern recognition (pp. 2536-2544).

---

### Official Review · Reviewer_DP9v · 2025-06-30

**Clarity:** 2
**Significance:** 2
**Originality:** 3
**Rating:** 4
**Confidence:** 4

**Summary:**

This paper proposes MAE-Pure, a novel adversarial purification defense method that denoises malicious inputs by restoring distorted semantic information while achieving state-of-the-art robust accuracy. The framework leverages the inherent sensitivity of Masked Autoencoders (MAE) to adversarial perturbations, framing purification as an optimization problem that minimizes reconstruction loss. This process works to reverse the Attention Matrix Variation (AMV) induced by the attack, effectively preserving the semantic relationships between image patches. To further bolster defense capabilities and model accuracy, the paper introduces fine-tuned variants like RMAE-Pure, which incorporate a classification loss to better preserve critical inter-patch semantic structures and significantly improve robustness.

**Questions:**

1. Can the MAE-Pure process be considered a form of "counter-adversarial" example against the original adversarial example, where its purpose is to decrease the MAE loss and consequently reduce the variation in the AMV?
2. The authors should focus on their writing, as the current version does not sufficiently highlight the paper's contributions.

**Ethical Concerns:**

["NO or VERY MINOR ethics concerns only"]

**Final Justification:**

The authors have addressed my concerns.

**Limitations:**

Yes

**Paper Formatting Concerns:**

There are no major formatting issues.

**Quality:**

2

**Strengths And Weaknesses:**

## Strengths

1. The paper introduces a novel and insightful approach to adversarial purification by focusing on preserving the semantic relationships between image patches.
2. The paper's claims are substantiated by an exceptionally thorough and rigorous set of experiments across multiple standard benchmarks.

## Weaknesses

1. The equivalence between the AMVM problem and the minimization of the MAE reconstruction loss, as presented, is not rigorously established. The paper posits that Equation (1) can be optimized by solving Equation (2), but the justification for this equivalence is insufficient. The authors should provide a more thorough theoretical explanation or proof of the conditions under which this equivalence holds.
2. The empirical validation presented in Figure 2 is not sufficiently convincing to support the associated claims. The visual differences in the attention weights between the clean, adversarial, and denoised examples are quite subtle. To strengthen the argument that adversarial perturbations distort semantic relations, the authors should enhance this visualization or provide additional quantitative analysis to more clearly and robustly demonstrate the claimed effect.
3. The paper claims to provide a defense against general adversarial attacks, but the methodology and experiments seem to focus primarily on a limited set of attacks, with AutoAttack being prominent. To validate the claim of general robustness, the experimental section should be expanded to include a wider and more diverse battery of adversarial attacks. If the method's effectiveness is indeed limited to specific types of attacks, this scope should be clearly articulated and discussed as a limitation.
4. Minor Issues: 1) Line 93: There is a placeholder note "(author?)" that needs to be addressed. 2) Line 36: There is a typo: "Differen" should be corrected to "Different". 3) Line 52: A comma is needed after "i.e.". The text should read "i.e., causing variation...".

---

> ### Author Rebuttal · Authors · 2025-07-31
>
> We sincerely thank the reviewer for the kind and constructive comments, which have greatly helped us improve the quality of our paper. We look forward to further engaging with you during the discussion phase.
>
> **Weakness 1:Equivalence between the AMVM problem and the minimization of the MAE reconstruction loss**
>
> Thank you for your valuable question regarding the optimization equivalence relationship between attention matrix variant (AMV) and the reconstruction loss.
> We believe that the equivalence between the attention matrix variant minimization (AMVM) problem and the minimization of the MAE reconstruction loss holds.
> We have demonstrated in **Theorem 3.1** and in Figure 4(a), (b), and (c) that adversarial perturbations lead to simultaneous increases in both the Attention Matrix Variation (AMV) and the MAE reconstruction loss. Moreover, as the magnitude of the adversarial perturbation increases, the trends of AMV and reconstruction loss consistently rise in a correlated manner.
> Furthermore, **Theorem 3.2** in our paper already provides a rigorous inequality and can be transformed into:
>
> $\frac{1}{2NT}\sum\_{t=1}^{T}\sum_{i=1}^{N} [HC\_A || \mathbf{A}^{\mathrm{dec}}\_{\mathrm{adv},i,t} -\mathbf{A}^{\mathrm{dec}}\_{i,t}||^2]\leq
> \mathcal{L}^{\mathrm{adv}}\_{\mathrm{rec}} - \frac{1}{2} \mathcal{L}\_{\mathrm{rec}}$
>
> which demonstrates that the AMV objective is upper-bounded by $\mathcal{L}^{\mathrm{adv}}\_{\mathrm{rec}}$. Since $\mathcal{L}\_{\mathrm{rec}}$ remains constant for a well-trained MAE.
> As $\mathcal{L}^{\mathrm{adv}}\_{\mathrm{rec}}$ decreases, the AMV also gradually decreases.
> Based on the above discussion, we therefore conclude that optimizing AMVM under the perturbation constraint $\Delta_\{\infty} \leq C\_e$ is equivalent to minimizing the reconstruction loss, namely
>
> $\min\_{\Delta}\mathcal{L}\_{\text{rec}}(x\_{\text{adv}}+\Delta)\Longleftrightarrow \min\_{\Delta}\mathcal{L}(\Delta) \qquad \text{s.t. } |\Delta|\_{\infty} \leq C\_e$
>
> where  $\min\_{\Delta}\mathcal{L}\_{\text{rec}}(x\_{\text{adv}} + \Delta)$  minimizes the MAE reconstruction loss for denoising, and $\min\_{\Delta} \mathcal{L}(\Delta) = \left\| \text{atten}(x\_{\text{adv}} + \Delta) - \text{atten}(x) \right\|\_2^2 $ minimizes the attention matrix variation to preserve inter-patch semantic consistency. **Theorem J.1** in the Appendix also ensures that optimizing Eq.(2) via PGD yields effective convergence, progressively guiding adversarial examples toward the semantic space of clean samples. This confirms the validity of Eq.(2) as a tractable and principled proxy for solving the original AMVM objective in Eq.(1).
> Moreover, the empirical analysis in Figure 4 (d) (e) and (f) further supports this theoretical derivation. The figure 4 (d),(e) and (f) shows that as we optimize $\min\_{\Delta}\mathcal{L}\_{\text{rec}}(x\_{\text{adv}}+\Delta)$, the reconstruction loss decreases, and the AMV decreases accordingly.
>
> Therefore, we believe our explanation of the equivalence between AMVM and the minimization of the MAE reconstruction loss is both sufficient and well-justified.
>
> **W2: Concerns about visualization**
>
> Thank you very much for this valuable comment. We appreciate your careful reading of our paper.
>
> Regarding the concern that *“the visual differences in the attention weights between the clean, adversarial, and denoised examples are quite subtle”*, we would like to clarify that these differences are actually **very evident** in Figure 2. For example, in the clean image shown in Figure 2(b), the target patch (highlighted by the red box) corresponds to a hole on the magnetic tape, and its most relevant image patch is another hole on the tape---this reflects a clear semantic relation. However, in the adversarial image shown in Figure~2(c), the same target patch’s most relevant patch is shifted to an unrelated edge region. This demonstrates that the MAE’s attention has been **significantly distorted by the adversarial perturbation**, which we believe makes the visualization itself quite convincing.
>
> In addition to this qualitative visualization, we also provide **quantitative evidence**: as shown in Figure~2(a), (b), and (c), we report that increasing the magnitude of adversarial perturbation leads to a measurable increase in the *attention matrix variation (AMV)*. This directly supports our claim that adversarial perturbations disrupt semantic relations as reflected in the attention mechanism.
>
> We hope this explanation clarifies why we believe the presented visualization, together with the quantitative AMV analysis, strongly supports our claim.
>
>
> **W3:  Limited attack evaluation**
>
> Thank you very much for this insightful comment. We sincerely appreciate your careful review and constructive suggestions.
>
> Regarding the concern that our methodology and experiments might focus on a limited set of attacks, we would like to clarify that our evaluation already covers a relatively broad range of adversarial attacks. Specifically, in Table 1, Table 2, and Table 4, we present results under PGD200+EOT20. In Table 3, we additionally report results under AutoAttack. Moreover, in Table 7, we further include evaluations with CW+EOT and DeepFool+EOT, and in Table 11, we provide results under BPDA.
>
> These experiments together demonstrate the robustness of our method under diverse attack strategies, not limited to AutoAttack. We hope this clarifies that our defense is not tailored to a single type of attack but has been validated across multiple strong and widely recognized adversarial attack methods.
>
> **W4: Typos correctness**
>
> Thank you very much for your kind correction of our typos. We will address and fix these errors in the revision.
>
> **Q1：Clarification on MAE-Pure as a Counter-Adversarial Process**
>
> Yes, your understanding is correct. The MAE-Pure process can indeed be considered a form of "counter-adversarial" transformation applied to adversarial examples. Its objective is to reduce the reconstruction loss (i.e., MAE loss), thereby suppressing the variation introduced in the AMV. By guiding the reconstruction towards the clean manifold, the purifier effectively neutralizes adversarial perturbations and stabilizes the downstream representation.
>
>
> **Q2: Clarifying and Emphasizing Our Contributions in the Revision**
>
> Thank you for your valuable feedback. We sincerely appreciate your suggestion and will revise the writing to better highlight the key contributions of our work.

---

> ### Author Response · Authors · 2025-08-04
>
> Dear Reviewer DP9v,
>
> I hope this message finds you well. As the discussion period is nearing its end with less than three days remaining, we would like to ensure that all your concerns have been addressed satisfactorily. If there are any additional points or feedback you would like us to consider, please don’t hesitate to let us know. Your insights are invaluable to us, and we are eager to resolve any remaining issues to further improve our work.
>
> Thank you again for your time and effort in reviewing our paper.
>
> Best
>
> Regard
>
> Team 21073

---

> > ### Comment · Reviewer_DP9v · 2025-08-05
> >
> > After reviewing the authors' rebuttal, I find that some concerns remain insufficiently addressed:
> >
> > 1. Equivalence justification. The reformulated inequality still only shows AMV is upper-bounded by L^{adv}_{rec}, not that minimizing reconstruction loss is equivalent to minimizing AMV. You fail to prove why optimizing one objective necessarily optimizes the other.
> >
> > 2. Theoretical rigor. You did not address the tightness of their bounds or the error analysis for multiple approximations (kernel approximation, Taylor expansion) used in their proofs, which is crucial for validating their theoretical claims.
> >
> > 3. Visualization clarity. I'm more concerned about how visually noticeable the changes are. You could use clearer and more distinct examples to help readers better understand how the adversarial perturbations affect the semantics.

---

> > > ### Author Response · Authors · 2025-08-09
> > > **Further inquire whether there are any remaining concerns**
> > >
> > > Dear Reviewer DP9v,
> > >
> > > As the author–reviewer discussion will conclude in less than 12 hours, we would like to sincerely thank you again for your time and thoughtful feedback.
> > >
> > > We believe that our previous responses have addressed all the concerns you raised. Before the discussion period ends, we would like to kindly check whether you have any new concerns, or if there are any parts of our replies that remain unclear. If so, we would be more than happy to provide further clarification.
> > >
> > > Thank you again for your engagement and constructive comments.
> > >
> > > Best regards,
> > >
> > > Your Friends
> > >
> > > All authors of Team 21073

---

> ### Author Response · Authors · 2025-08-05
>
> Dear Reviewer DP9v
>
> We sincerely appreciate your response and are pleased that we were able to address some of your concerns. Moving forward, we hope that further discussion will help clarify any remaining questions you may have.
>
>
> **For Equivalence**
>
> We would like to clarify that our formulation is mathematically equivalent in the strict sense. As you rightly noted, AMV is upper-bounded by $\mathcal{L}^{\text{adv}}\_{\text{rec}}$, which implies that as $\mathcal{L}^{\text{adv}}\_{\text{rec}} - \frac{1}{2} \mathcal{L}\_{\text{rec}}\rightarrow 0$, AMV will also converge to zero, because AMV is non-negative.
>
> **Furthermore, we would like to clarify that minimizing an upper bound to indirectly optimize an intractable or non-differentiable objective is a well-established and widely accepted practice in the machine learning and optimization community [1,2,3,4]. We thus hope this clarification helps reconcile the concern regarding the optimization objective.**
>
> **For the Sake of Rigor**
>
> We would like to clarify that the approximations used in our theoretical analysis, particularly the kernel approximation and the Taylor expansion, are not arbitrary choices. Rather, they are well-established and widely adopted techniques in the machine learning and optimization communities. These methods have been extensively employed in numerous prior works, especially in contexts such as [5,6,7,8], and have been repeatedly validated in influential papers published in top-tier journals and conferences. We emphasize that our use of these techniques follows standard practices in the field and builds upon the stability and guarantees established in the existing literature. **Therefore, we are confident that our analysis adheres to the rigorous standards upheld by prior works in top-tier venues, and we welcome any suggestions for further clarification or strengthening.**
>
> **For Visualization**
>
> We present three sets of examples in our paper: the first is shown in Figure 2, while the second and third are included in Figure 8 of the appendix. In particular, the second row of Figure 8 clearly illustrates how the adversarial noise affects the semantic content. This example was carefully chosen to make the semantic impact of the perturbation visually salient and intuitive.
>
> We sincerely appreciate your response once again. Should there be any further questions, we would be happy to engage in continued discussion.
>
> Best Regards
>
> Your Friends
>
> Team 21073
>
>
> [1] Hinton, G., Vinyals, O., & Dean, J. (2015). Distilling the knowledge in a neural network.  NIPS 2017
>
> [2] Wong, E., & Kolter, Z. (2018, July). Provable defenses against adversarial examples via the convex outer adversarial polytope. In International conference on machine learning (pp. 5286-5295). PMLR.
>
> [3] Bartlett, P. L., Jordan, M. I., & McAuliffe, J. D. (2006). Convexity, classification, and risk bounds. Journal of the American Statistical Association, 101(473), 138-156.
>
> [4] Zhang, S., Qian, Z., Huang, K., Wang, Q., Zhang, R., & Yi, X. (2021, July). Towards better robust generalization with shift consistency regularization. In International conference on machine learning (pp. 12524-12534). PMLR.
>
> [5] Moosavi-Dezfooli, S. M., Fawzi, A., & Frossard, P. (2016). Deepfool: a simple and accurate method to fool deep neural networks. In Proceedings of the IEEE conference on computer vision and pattern recognition (pp. 2574-2582).
>
> [6] Neyshabur, B., Bhojanapalli, S., McAllester, D., & Srebro, N. (2017). Exploring generalization in deep learning. Advances in neural information processing systems, 30.
>
> [7] Choromanski, K., Likhosherstov, V., Dohan, D., Song, X., Gane, A., Sarlos, T., ... & Weller, A. (2020). Rethinking attention with performers. NeurIPS 2021.
>
> [8] Nguyen, T., Pham, M., Nguyen, T., Nguyen, K., Osher, S., & Ho, N. (2022). Fourierformer: Transformer meets generalized fourier integral theorem. Advances in Neural Information Processing Systems, 35, 29319-29335.

---

> ### Author Response · Authors · 2025-08-07
> **Invitation for Further Discussion and Clarification**
>
> Dear reviewer DP9v
>
> Thank you very much for your kind and constructive review.
>
> Regarding your previous concerns on Equivalence Justification, Theoretical Rigor, and Visualization Clarity, we believe we have now addressed them thoroughly in our discussion.
>
> For Equivalence Justification and Theoretical Rigor, our formulation strictly follows the methodology adopted in several influential prior works as shown in last discussion. These references provide the theoretical foundation for our approach, and we have ensured that our derivation remains faithful to the rigor established in these studies.
>
> For Visualization Clarity, we have shown more illustrative examples in Figure 8, which we hope make the semantic effects of adversarial perturbations more intuitively observable.
>
> Please let us know if there are any remaining or new concerns—we would be more than happy to clarify or further elaborate.
>
> Warm regards,
>
> Your Friends,
>
> Team 21073

---

### Official Review · Reviewer_tuPn · 2025-06-30

**Clarity:** 2
**Significance:** 3
**Originality:** 3
**Rating:** 4
**Confidence:** 3

**Summary:**

The paper observes that MAEs are highly sensitive to imperceptible adversarial perturbations, which severely disrupt the semantic relationships between image patches. Unlike existing methods relying on powerful generative models like diffusion models, this work focuses on preserving these inter-patch semantic relationships. It transforms purification into an optimization problem of Attention Matrix Variation (AMV) and designs MAEPure to achieve SOTA performance.

**Questions:**

1. Although the problem and motivation of MAE as a purifier are reasonably explained in the Introduction, the deep motivation for using MAE compared to the successful purifier of the diffusion model is not reasonably explained.

2. In Table 10, why does the proposed method usually take more time on CIFAR10 but less time on ImageNet than DiffPure?

3. It is recommended to add comparison with other papers that use MAE as a purifier.

  [1] MAEDefense: An Effective Masked AutoEncoder Defense against Adversarial Attacks

  [2] Adversarial Masked Autoencoder Purifier with Defense Transferability

4. In Theorem 3.2, the author proposed a lower bound of L_{rec}^{adv}. But if it is used as a guide for the construction of the objective about attention in Section 4.1, I think it is necessary to find the upper bound of L_{rec}^{adv}, because when we minimize the upper bound, we can effectively reduce L_{rec}^{adv}.

5. It is recommended to add explanations to Figure 3 to help readers understand the meaning of the figure.

6. In the introduction to MAE in Section L of the supplementary material, it is recommended to add the description of 'Large-MAE' and 'the author (line 789)', such as citations, to help readers reproduce or understand.

7. There is a coding problem on line 93: milestone study (author?), please check.

**Ethical Concerns:**

["NO or VERY MINOR ethics concerns only"]

**Final Justification:**

My questions have been solved by authors.

**Limitations:**

yes

**Quality:**

3

**Strengths And Weaknesses:**

Strengths
1. The authors provide a sufficient subjective presentation of the problems and solutions faced when using MAE, which enables readers to understand more clearly.
2. The authors provide sufficient theoretical explanations for the effectiveness of MAE and the selection of AMV to make the method more solid.

Weaknesses
1. The authors did not clearly present the core motivations and advantages of choosing MAE.
2. Some additional explanations are needed in the methods and experiments

---

> ### Author Rebuttal · Authors · 2025-07-31
>
> We sincerely thank the reviewer for the kind and constructive comments, which have greatly helped us improve the quality of our paper. We look forward to further engaging with you during the discussion phase.
>
>
> **W1 and W2**
>
> Thank you very much for your valuable comment, and we sincerely apologize for not having made our motivation clear in the original manuscript.
>
> Our motivation for choosing the MAE architecture is based on its observed sensitivity to adversarial perturbations when using attention mechanism for reconstruction, as shown in Figure 1. We further investigated the root cause and found that this sensitivity originates from the *attention matrix* in MAE, which is highly susceptible to noise. This is theoretically justified in Theorem 3.1 and empirically validated in Figure 3 (a), (b), and (c).
>
> Based on this phenomenon, we propose MAE-Pure, which aims to suppress adversarial noise by constraining the variation of the attention matrix (AMV). This mechanism is grounded in the proven consistency between AMV and reconstruction loss, as established in Theorem 3.2 and further supported by Figure 3 (d), (e), and (f). Therefore, by optimizing the reconstruction loss, MAE-Pure effectively reduces AMV, thus achieving denoising.
>
> Furthermore, in Figure 7, we demonstrate that the same characteristics hold for masked diffusion-based model, e.g., MaskDiT. Leveraging these insights, we integrated our method into MaskDiT and observed consistent and strong performance.
>
> **Q1: Selection for MAE**
>
> Thank you for your valuable comment. While diffusion-based models have recently demonstrated strong performance in purification tasks, our motivation for using MAE based on two reasons:
> (1) To identify the purification effectiveness is solely produced by minimizing *Attention Matrix Variation (AMV)*, we choose the MAE which only includes the attention mechanism. (2) To generalize our observation to a more complex model architecture, we integrate our AMV minimization into MaskDiT that not only includes the attention mechanism but also embodies powerful generation capability.
>
> In our paper, we provide both empirical and theoretical evidence for the effectiveness of MAE as a purifier. Specifically, in **Figure 4(a), (b), and (c)** and **Theorem 3.1**, we show that adversarial perturbations lead to significant variations in the attention matrices of MAE. These variations are systematically aligned with the adversarial noise, revealing an intrinsic vulnerability.
>
> Building upon this, we introduce the concept of AMV as a quantitative measure of perturbation sensitivity. Furthermore, in **Figure 4(d), (e), and (f)** and **Theorem 3.2**, we demonstrate that minimizing AMV provides a principled and effective strategy for denoising adversarial noises.
>
> More importantly, our insights are not limited to MAE. As shown in **Figure 7**, the AMV-based purification principle can be seamlessly extended to MaskDiT leading to our proposed method **MaskDiT-Pure**. By leveraging the strong generative capability of diffusion models, MaskDiT-Pure achieves superior performance compared to existing other diffusion-based purification baselines.
>
> In summary, our use of MAE is not only empirically validated but also theoretically motivated, and the proposed framework generalizes beyond MAE to improve even diffusion-based purifiers.
>
> **Q2: Concerns about runtime**
>
> Thank you for your question. The observed runtime difference stems from the fact that DiffPure employs different diffusion models on CIFAR-10 and ImageNet. Specifically, for CIFAR-10, DiffPure uses a score-based SDE model, while for ImageNet, it adopts a Guided Diffusion.
>
> Notably, guided diffusion is known for its stronger generative capacity and better sample quality, but it also incurs higher computational overhead compared with score-based diffusion. As a result, even excluding the influence of input size, DiffPure takes significantly more time on ImageNet than on CIFAR-10, primarily due to the heavier computational cost of the guided diffusion process.
>
> To summarize, the difference in runtime patterns mainly arises from DiffPure's use of different diffusion models, whereas our method ensures consistency by design. This also explains why our method shows the higher runtime on CIFAR-10 but the lower runtime on ImageNet compared to DiffPure.
>
> **Q3: Comparison with more different baselines**
>
> Thank you for the constructive suggestion. Following your recommendation, we have included a comparison with two recent works that adopt Masked Autoencoders (MAE) as purifiers: MAEDefense [1] and MAEP [2].
> %
> We reproduced their methods on both CIFAR-10 and SVHN datasets under a strong adaptive attack setting using PGD-200 + EOT-20 with full gradient backpropagation, which aligns with the evaluation protocol used in our paper. The backbone classifier for all methods is WideResNet-28-10 for a fair comparison.
>
> The results are summarized below:
>
> **Table: Comparison with new MAE baseline on CIFAR10 and SVHN**
>
> | Method           | Std Acc | Robust Acc (ℓ∞) | Robust Acc (ℓ2) | Std Acc | Robust Acc (ℓ∞) | Robust Acc (ℓ2) |
> |------------------|---------|------------------|------------------|---------|------------------|------------------|
> |                  | **CIFAR10**             |                  |                  | **SVHN**             |                  |                  |
> | MAEDefense       | 88.17   | 29.29           | 44.15           | 90.50   | 34.15           | 49.28           |
> | MAEP             | 92.30   | 32.51           | 52.15           | 91.33   | 32.59           | 50.58           |
> | MAE-Pure         | 88.57   | 40.53           | 53.50           | 94.54   | 27.59           | 55.29           |
> | MaskDiT-Pure     | 92.03   | 50.57           | 64.53           | 94.91   | 46.57           | 66.38           |
> | RMAE-Pure        | 90.09   | 45.15           | 60.72           | 94.47   | 39.15           | 60.51           |
> | **RMaskDiT-Pure**| **93.11** | **62.13**     | **73.57**       | **95.39** | **55.90**       | **70.18**       |
>
> We will add the results in our revision.
>
> **Q4: Optimaztion for upper bound**
>
> Thank you for your insightful comment.
> %
> We agree that minimizing an upper bound can effectively reduce the target loss, such as $ \mathcal{L}\_{\text{rec}}^{\text{adv}} $. However, our method does not involve constructing or optimizing any upper bound of $\mathcal{L}\_{\text{rec}}^{\text{adv}} $; instead, we directly minimize $\mathcal{L}\_{\text{rec}}^{\text{adv}} $ itself by iterative update the adversarial examples for purification.
> Therefore, finding an upper bound does not provide direct benefit to our current approach.
>
>
> Moreover, please see the reformulated inequality of Theorem 3.2 in response for Reviewer DP9v.
> We would like to clarify that the purpose of Theorem~3.2 is not to establish a lower bound of $ \mathcal{L}\_{\text{rec}}^{\text{adv}}$, but rather to show that the attention matrix variation (AMV) term, i.e., $
> \sum\_{t=1}^{T}\sum\_{i=1}^{N}\left\| \mathbf{A}^{\mathrm{dec}}\_{\mathrm{adv},i,t} - \mathbf{A}^{\mathrm{dec}}\_{i,t} \right\|^2,$
> can be upper-bounded by $\mathcal{L}\_{\text{rec}}^{\text{adv}}$.
>
> This holds because the reconstruction loss of clean samples remains approximately constant for a well-trained MAE.
> As a result, minimizing $\mathcal{L}\_{\text{rec}}^{\text{adv}}$ implicitly reduces AMV, which helps preserve the inter-patch semantic relationships and thus achieves denoising.
>
> **Q5: Exploration of Figure 3**
>
> Thank you for the suggestion. We have added detailed explanations to Figure 3 to clarify the denoising pipeline of MAE-Pure.
>
> Specifically, Figure 3 provides an overview of the proposed purification framework. It illustrates the iterative denoising process applied to adversarial examples. Starting from an adversarial input $x^{\text{adv}}$, the model applies Projected Gradient Descent (PGD) iterations, where at each step $s$, the reconstruction loss of the MAE is minimized to generate a purified sample $x^{\text{adv}}\_s$. This iterative process aims to reduce the Attention Matrix Variation (AMV) by aligning the attention distributions of adversarial and clean inputs. The figure also depicts the relationship between the MAE reconstruction loss and the adversarial perturbations being gradually removed, ultimately yielding a semantically faithful, purified image. The shaded areas and clip operations represent the bounded perturbation constraints ($\| \Delta \|\_{\infty} \leq C\_e$) used during optimization.
>
> We will include the improved explanation in the revision.
>
> **Q6: Introduction of Large MAE**
>
> Thank you for your helpful comment. The term *Large-MAE*  refers to the ViT-Large architecture introduced in the original MAE paper [3]. We will add the appropriate citation and clarify this description in Section~L of the supplementary material in the revision to improve reproducibility and reader understanding.
>
> **Q7: Coding problem**
>
> Thank you for your kind suggestion. We will address this issue in the revision.
>
> [1] Lyu, W., Wu, M., Yin, Z., & Luo, B. (2023, October). Maedefense: An effective masked autoencoder defense against adversarial attacks. In 2023 Asia Pacific Signal and Information Processing Association Annual Summit and Conference (APSIPA ASC) (pp. 1915-1922). IEEE.
>
> [2] Chen, Y. C., & Lu, C. S. (2025). Adversarial Masked Autoencoder Purifier with Defense Transferability. arXiv preprint arXiv:2501.16904.
>
> [3] He, K., Chen, X., Xie, S., Li, Y., Dollár, P., & Girshick, R. (2022). Masked autoencoders are scalable vision learners. In Proceedings of the IEEE/CVF conference on computer vision and pattern recognition (pp. 16000-16009).

---

> > ### Comment · Reviewer_tuPn · 2025-08-01
> >
> > Thanks to the authors' detailed responses, my questions have been partially addressed, but I still have some questions:
> >
> > For Q1: Sorry for the lack of clarity in my previous review. I'd rather understand the motivation behind the problem, rather than the motivation behind MAE. For example, what core problem is currently encountered when using diffusion as a purifier that warrants the use of MAE, rather than the motivation behind using MAE because of its advantages or some properties.
> >
> > For Q2: In my opinion, MAE is a much lighter model compared to the diffusion model. Why do we still pay such a high time cost when using it?
> >
> > For Q5: I'm still a little confused by the authors' explanation, so I'd like to use a simple example to illustrate my point. I want to minimize a, so I monitor its upper bound. When it is limited to below 2, then a must be less than 2. But if I monitor its lower bound, even if it is limited to below 2, a may still be 100. I'm not sure if there is something I still don't understand correctly.

---

> ### Author Response · Authors · 2025-08-01
>
> Dear Reviewer tuPn
>
> We sincerely appreciate your positive and friendly response, and we are also glad to have resolved some of your concerns. For your remaining questions, we are more than willing to engage in further communication to provide clarification.
>
> **For Q1:**
> We thank the reviewer for the helpful clarification. We understand that the concern lies in the core problem encountered when using diffusion as a purifier, rather than the motivation for using MAE per se. Existing diffusion-based methods, such as DiffPure aim to project adversarial examples back toward the clean data distribution. However, they tend to overlook the analysis and optimization of semantic dependencies between image patches (i.e., inter-patch relationships), which are critical for effective denoising.
> Through our empirical analysis, we observe that inter-patch relationships are particularly sensitive to adversarial noise. This suggests, from an intuitive perspective, that explicitly modeling and preserving these dependencies could lead to better purification outcomes.
> **Importantly, our proposed mechanism is not in conflict with diffusion-based approaches. Rather, it represents a general principle that can be instantiated in different architectures, including autoencoder-based models (e.g., MAE) and diffusion-based models (e.g., MaskDiT). Our results demonstrate that incorporating this principle consistently enhances denoising performance across both types.**
>
> **For Q2:** We agree with the reviewer that MAE is generally a lighter model compared to diffusion-based models. However, in our setting, we apply the MAE-based denoising process iteratively over multiple steps to achieve strong purification performance. This iterative process significantly increases the overall runtime, which explains the relatively high time cost despite the lightweight nature of the MAE architecture.
>
> **For Q4**: Thank you for your insightful comment. To clarify, our objective is to minimize the term
> $
> \sum_{t=1}^{T}\sum\_{i=1}^{N}\left\| \mathbf{A}^{\mathrm{dec}}\_{\mathrm{adv},i,t} - \mathbf{A}^{\mathrm{dec}}\_{i,t} \right\|^2,
> $
> which measures the distortion in the decoded attention maps under adversarial perturbation. We achieve this by monitoring and optimizing its upper bound, specifically
> $
> \mathcal{L}^{\mathrm{adv}}\_{\mathrm{rec}} - \tfrac{1}{2} \mathcal{L}\_{\mathrm{rec}},
> $
> where $\mathcal{L}\_{\mathrm{rec}}$ is treated as a constant for a well-trained MAE.  This approach allows us to indirectly constrain the adversarial distortion via an analytically tractable surrogate.
>
> In addition, we also agree with your statement regarding obtaining an upper bound of the loss $\mathcal{L}^{\mathrm{adv}}\_{\mathrm{rec}}$. Therefore, we derive the loss upper bound as follows.
> $\mathcal{L}^{\mathrm{adv}}\_{\mathrm{rec}} \leq \mathcal{L}\_{\mathrm{rec}} + \frac{1}{(n-m)}\left| (L^2+1)||\delta||^2 \right|$,
> where $\mathcal{L}$ is the Lipschitz constant of the MAE. $N$ represents the total number of samples, $n$ denotes the number of patches per sample, $m$ refers to the number of masked patches, and $\delta$ represents the perturbation. The inequality shows that, under the given MAE architecture and data, the adversarial reconstruction loss $\mathcal{L}^{\mathrm{adv}}\_{\mathrm{rec}}$ is determined by the perturbation $\delta$. Since $\delta$ is typically small, this implies that the upper bound of $\mathcal{L}^{\mathrm{adv}}\_{\mathrm{rec}}$ remains controllable.
>
>
> The proof is as follows:
>
> Let $x\_{i,1}$ denote the unmasked (visible) part of the $i$-th sample, and $x\_{i,2}$ denote the masked (invisible) part of the $i$-th sample, consistent with the notation used in the main paper.
>
>
>
>
> $\mathcal{L}^{\mathrm{adv}}\_{\mathrm{rec}}  = \frac{1}{N(n-m)}\sum_{i=1}^{N}||g(f(x\_{i,1}^{adv}))-x\_{i,2}^{adv}||^2
> $
>
> $
> = \frac{1}{N(n-m)}\sum_{i=1}^{N}\left|||g(f(x\_{i,1}^{adv}))-x\_{i,2}^{adv} + g(f(x\_{i,1}))+x_{i,2} -  g(f(x\_{i,1}))-x\_{i,2} ||^2\right|$
>
> $
> \leq \frac{1}{N(n-m)}\sum_{i=1}^{N}\left|||g(f(x\_{i,1}^{adv}))- g(f(x\_{i,1})) ||^2 + ||g(f(x\_{i,1})) - x\_{i,2} ||^2 + ||x\_{i,2}^{adv} - x\_{i,2}||^2\right|$
>
> $\leq\frac{1}{N(n-m)}\sum\_{i=1}^{N}||g(f(x\_{i,1})) - x\_{i,2} ||^2+ \frac{1}{N(n-m)}\sum_{i=1}^{N}\left| ||x\_{i,2}^{adv}-x\_{i,2}||^2 + ||g(f(x\_{i,1}^{adv})) - g(f(x\_{i,1}))||^2\right|$
>
> Based on a widely used assumption that $||g(f(a))- g(f(b))||  \leq L||a-b||$, we can get
>
> $\leq\mathcal{L}\_{\mathrm{rec}}+ \frac{1}{N(n-m)}\sum\_{i=1}^{N}\left| ||x\_{i,2}^{adv}-x\_{i,2}||^2 + L^2||\delta||^2\right|$
>
> $\leq \mathcal{L}_{\mathrm{rec}} + \frac{1}{(n-m)}\left| (L^2+1)||\delta||^2 \right|$
>
> We sincerely thank the reviewer for the professional and constructive comments. These suggestions have greatly improved the quality of our paper. We will incorporate all the points you mentioned into our revision.
>
> Best Regards,
>
> Your friends,
>
> Team 21073

---

> > ### Comment · Reviewer_tuPn · 2025-08-05
> >
> > Thank you for your detailed response. I will improve my score from 'borderline reject' to 'borderline accept' concerning the additional responses. However, I still have a minor question to discuss with the authors: Regarding Q2, as MAE itself is relatively lightweight, and I believe it has the potential to solve the current problem of very high purification computational complexity. I briefly experimented with a small amount of MAE and AT based on the framework provided by the authors. It works with very low attack steps, but it degrades significantly when attack steps increase. I suspect this may be related to the low randomness caused by the low number of cascaded MAEs. However, I still believe that the potential of MAE in purifiers has not been fully explored, and I would like to hear the authors' thoughts.

---

> > > ### Author Response · Authors · 2025-08-05
> > > **Thank You for Your Thoughtful Feedback and Score Adjustment**
> > >
> > > Dear Reviewer tuPn,
> > >
> > > First of all, we sincerely appreciate your willingness to raise the score and your recognition of our work. Thank you also for sharing your insightful thoughts and preliminary experiments.
> > >
> > > Regarding your question:
> > >
> > > We agree with your observation that MAE is inherently lightweight and holds great potential for addressing the high computational complexity of purification. We are also eager to explore this direction further in our future work.
> > >
> > > We acknowledge that the low randomness caused by a limited number of cascaded MAEs may indeed contribute to the performance degradation under stronger attacks. Additionally, we believe this issue is closely tied to the generative capacity of MAE itself. This is precisely why we introduce MaskDiT, which aims to provide stronger generative robustness compared to standard MAEs.
> > >
> > > We also fully agree that the potential of MAE-based purifiers has not yet been fully explored. A key factor is that transformer-based models—whether for classification or generation—are typically data-hungry. With small datasets such as CIFAR-10, it is often difficult to fully unleash their capacity. To this end, we believe that data augmentation strategies, such as diffusion-based data augmentation, could further enhance their generative ability and robustness.
> > >
> > > Once again, thank you very much for your thoughtful question and encouraging feedback.
> > >
> > > Best regards,
> > >
> > > Your friends,
> > >
> > > Team 21073

---

> ### Author Response · Authors · 2025-08-04
>
> Dear Reviewer tuPn,
>
> I hope this message finds you well. As the discussion period is nearing its end with less than three days remaining, we would like to ensure that all your concerns have been addressed satisfactorily. If there are any additional points or feedback you would like us to consider, please don’t hesitate to let us know. Your insights are invaluable to us, and we are eager to resolve any remaining issues to further improve our work.
>
> Thank you again for your time and effort in reviewing our paper.
>
> Best
>
> Regard
>
> Team 21073

---

### Official Review · Reviewer_eQ1q · 2025-07-03

**Clarity:** 3
**Significance:** 2
**Originality:** 2
**Rating:** 4
**Confidence:** 3

**Summary:**

The paper proposes MAE-Pure, a new adversarial purification method that leverages Masked Autoencoder (MAE) to purify adversarial examples. MAE-Pure is motivated by the observation that adversarial noise significantly degrades the MAE’s ability to reconstruct images by distorting the semantic relationships among patches. MAE-Pure minimizes the variation in the attention matrix and iteratively reduces adversarial perturbations during the training. MAE-Pure can be extended to MaskDiT as well. Experiments show that MAE-Pure outperforms baseline methods.

**Questions:**

1. RMaskDiT-Pure/MaskDiT-Pure outperforms RMAE-Pure/MAE-Pure by a notable margin in Table 1. I am wondering if it is the idea itself boosting the performance of RMastDiT-Pure/MaskDiT-Pure, or it is because the RMaskDiT/MaskDiT model itself is strong at adversarial purification? I hope authors could add more experiments to clarify this (e.g., an ablation study on MaskDiT and the proposed MaskDiT-Pure).
2. Following Q1, how does MaskDiT perform in the motivation experiments? Can the motivation results still hold for MaskDiT?
3. Just for curiosity, for the empirical validation, why are the selected datasets inconsistent? For example, SVHN is selected for investigating how perturbation influences reconstruction, while ImageNet is selected for how semantic relationship between different patches changes under perturbations. Could the authors clarify on this?
4. Since the robust purification model requires further fine-tuning process using classification loss, I am curious about how the proposed methods perform if the downstream classifiers are changed into ViT-based architectures? It seems that the authors only use ResNet-based architectures for experiments.

**Ethical Concerns:**

["NO or VERY MINOR ethics concerns only"]

**Final Justification:**

After reading the rebuttal, most of my concerns are addressed, so I will increase my score to 4.

**Limitations:**

The authors include a short section that discusses one limitation of the proposed method in the Appendix. However, the authors did not include future work to address this limitation.

**Quality:**

2

**Strengths And Weaknesses:**

**Strengths**

- This paper is easy to follow and the idea is simple.
- This paper is theoretically motivated with sufficient justifications.
- Experiments are solid and sufficient, although the standard deviations are not reported.

**Weaknesses**

- Standard deviations are not reported, so the stability of MAE-Pure needs to be verified.
- Missing citation in line 93 (minor issue).
- Although the authors mention the type of attacks used for evaluation, the threat model remains unclear. For example, in Table 2, ‘the experiment are implemented on WideResNet-28-10’. Does that mean the threat model is WRN-28-10? Or, is it (Purification model + WideResNet-28-10)? More clarifications are needed since different attack settings can largely affect the robustness.
- The training/fine-tuning cost of the proposed methods is not reported. It would add more value to this paper if the training/fine-tuning process of MAE and MaskDiT is also efficient.

---

> ### Author Rebuttal · Authors · 2025-07-31
>
> We sincerely thank the reviewer for the kind and constructive comments, which have greatly helped us improve the quality of our paper. We look forward to further engaging with you during the discussion phase.
>
>
> **Weakness 1: Standard deviations**
>
> Thank you for the reviewer’s kind suggestion. We have included the standard deviations for CIFAR-10, CIFAR-100 here SVHN.
> The results are obtained from 5 random runs.
>
>
> **Table: Results on CIFAR10.**
>
> | Method         | Std Acc           | ℓ∞ Robust Acc       | ℓ2 Robust Acc       |
> |----------------|-------------------|----------------------|----------------------|
> | MAE-Pure       | 88.57 ± 1.2       | 40.53 ± 1.8          | 53.50 ± 2.0          |
> | MaskDiT-Pure   | 92.03 ± 1.0       | 50.57 ± 1.7          | 64.53 ± 2.1          |
> | RMAE-Pure      | 90.09 ± 0.9       | 45.15 ± 1.4          | 60.72 ± 1.9          |
> | RMaskDiT-Pure  | **93.11 ± 0.8**       | **62.13 ± 1.6**      | **73.57 ± 2.2**      |
>
> **Table: Results on CIFAR100 and SVHN (left side is CIFAR100 and right side is SVHN).**
>
> | Method         | CIFAR-100 Std Acc | ℓ∞ Robust Acc | ℓ2 Robust Acc | SVHN Std Acc   | ℓ∞ Robust Acc | ℓ2 Robust Acc |
> |----------------|-------------------|----------------|----------------|----------------|----------------|----------------|
> | MAE-Pure       | 65.34 ± 1.8       | 14.28 ± 1.3     | 29.29 ± 2.1     | 94.54 ± 1.5     | 27.59 ± 1.2     | 55.29 ± 2.3     |
> | MaskDiT-Pure   | **70.03 ± 1.2**       | 24.39 ± 1.6     | 36.51 ± 2.0     | 94.91 ± 0.9     | 46.57 ± 2.1     | 66.38 ± 2.5     |
> | RMAE-Pure      | 66.28 ± 1.4       | 19.53 ± 1.5     | 31.58 ± 1.9     | 94.47 ± 1.1     | 39.15 ± 1.8     | 60.51 ± 2.0     |
> | RMaskDiT-Pure  | 69.87 ± 1.0       | **29.91 ± 1.4**     | **43.27 ± 2.3**     | **95.39 ± 1.3**     | **55.90 ± 1.7**     | **70.18 ± 2.4**     |
>
>
>
>
>
> **Weakness 2: Missing citation**
>
> Thank you for pointing this out. We will address the missing citation in line 93 and ensure it is properly included in the revision.
>
> **Weakness 3 :Concerns or clarifications regarding the threat model**
>
> We appreciate the reviewer’s insightful comment.
> In our experiments, the threat model is indeed the purification model combined with a classifier (e.g. MaskDiT + WRN28).
> This setting is consistent with prior works [1,2], and we adopted the same configuration to ensure a fair and direct comparison.
> %
> We will make a more clear exploration in the revision.
>
> **W4: Cost of different method**
>
> We presented the training time of the MAE-Pure/MaskDiT-Pure purifier and compared it with the DiffPure purifier. At the same time, we also performed a finetuning time comparison between RMAE-Pure/RMaskDiT-Pure and ADDT. We used 8* A40 GPUs for parallel training with a batch size of 128.
>
> **Table: Time consumption for model training (in hours).**
>
> | Dataset   | Model     | Time (h) |
> |-----------|-----------|----------|
> | CIFAR-10  | MAE       | **7.2**      |
> |           | MaskDiT   | 8.9      |
> |           | Diffpure  | 12.4     |
> | ImageNet  | MAE       | **19.7**     |
> |           | MaskDiT   | 21.9     |
> |           | Diffpure  | 29.2     |
>
> Next table is finetuning time.
>
> **Table: Time consumption for robust model fine-tuning (in hours).**
>
> | Dataset   | Model     | Time (h) |
> |-----------|-----------|----------|
> | CIFAR-10  | RMAE       | **1.4**      |
> |           | RMaskDiT   | 2.9      |
> |           | ADDT      | 6.5      |
> | ImageNet  | RMAE       | **22.5**     |
> |           | RMaskDiT   | 52.3     |
> |           | ADDT  | 65.0     |
>
> **Q1: Ablation on the Effectiveness of MaskDiT framework**
>
> Thank you for your thoughtful question about whether the performance gain comes from the proposed Pure framework or the backbone architecture itself.
> We have already conducted ablation studies to address this concern in the submitted Supplementary Material. Please note that this part is not included in the Appendix. We present the key results in Supplementary Material below for clarity:
>
> **Table: Ablation analysis on different denoising components across various datasets**
>
> | Method                      | Architecture | CIFAR10 Std Acc | CIFAR10 Robust Acc | CIFAR100 Std Acc | CIFAR100 Robust Acc | SVHN Std Acc | SVHN Robust Acc |
> |----------------------------|--------------|------------------|---------------------|-------------------|----------------------|---------------|------------------|
> | MaskDiT$\_{\text{/purification}}$  | WRN-28-10     | 91.13            | 42.99               | 63.29             | 13.57                | 92.28         | 40.07            |
> | RMaskDiT$\_{\text{/purification}}$ | WRN-28-10     | 90.57            | 47.57               | 64.15             | 18.55                | 93.03         | 45.19            |
> | **MaskDiT-Pure**           | **WRN-28-10** | 92.03        | 50.57           | 70.03         | 24.39            | 94.91     | 46.57        |
> | **RMaskDiT-Pure**          | **WRN-28-10** | **93.11**        | **62.13**           | **69.87**         | **29.91**            | **95.39**     | **55.90**        |
>
> We begin with training two models, e.g., MaskDiT and RMaskDiT, used for purification.
> Firstly, following the DiffPure [3], we conduct the reconstruction-based purification processes for purification, which are MaskDiT$_{\text{/purification}}$ and RMaskDiT$\_{\text{/purification}}$. This reconstruction based purification involves a forward (noise addition) and backward (denoising) pass. The reconstructed images are treated as purified results to investigate whether the superior performance stems from generative ability of selected models.
> On the other hand, we then compared the above performances with the proposed AMV-based purification methods by using the same models, where the purification processes are named as MaskDiT-Pure and RMaskDiT-Pure.
> By keeping the model architecture consistent across all variants, we enabled a direct comparison of the effectiveness of different denoising strategies. **It is note that MaskDiT and RMaskDiT indeed possess a certain capability to withstand stronger adversarial attacks; however, their effectiveness is still much lower than that of our proposed MaskDiT-Pure and RMaskDiT-Pure.**
>
> **Q2: Motivation experiments on MaskDiT**
>
> We assume that by "motivation experiments" you are referring to Figure 2, which illustrates the impact of adversarial perturbations on the attention matrix in MAE.
> We believe that the motivations and observations from these experiments are consistent and applicable to MaskDiT.
>
> To support the above conclusion, we have conducted the quantitative analysis about the relationship among the adversarial perturbations, reconstruction loss, and attention matrix variance (AMV) in Figure 7.
> The results demonstrate that MaskDiT exhibits a similar pattern with MAE, where adversarial perturbations lead to noticeable changes in attention variance and increased reconstruction loss.
> This consistency suggests that the motivation, namely that adversarial noise disrupts attention-based semantic representations, also applies to MaskDiT.
> Therefore, we conclude that the motivation remains valid for MaskDiT based on our empirical evidence.
>
> **Q3: Justification for Dataset Selection in Motivation Experiments**
>
> We appreciate the reviewer’s thoughtful question. The datasets were intentionally chosen to align with the specific phenomena we aimed to highlight in different parts of our empirical analysis.
> In Figure 1, SVHN was selected to study the **impact of adversarial perturbations on MAE’s reconstruction quality**. SVHN images are relatively simple and contain less semantic variation, making the degradation caused by adversarial noise more visually apparent and interpretable. This helps clearly demonstrate reconstruction failure under perturbations.
> In contrast, Figure 2 focuses on analyzing **how adversarial perturbations disrupt the semantic relationships** between image patches. For this purpose, we used ImageNet, as its images contain richer semantics, diverse objects, and complex spatial structures. This allows us to more convincingly showcase the disruption in inter-patch attention and the resulting semantic inconsistency.
>
> **Q4: ViT-based method**
>
> Thank you for the reviewer’s question and suggestion. Indeed, in our current experiments, we have only used ResNet-based architectures to validate the effectiveness of the robust purification model. To address the reviewer's suggestion regarding the use of ViT-based architectures (e.g., ViT-B), we plan to conduct further experiments on the CIFAR-10 dataset and use the ViT-B (Vision Transformer) model as the downstream classifier to evaluate the performance of our proposed method under this architecture. We use PGD200+EOT20 with full gradient aligned with main paper.
>
> **Table: Clean and robust accuracy (%) on CIFAR-10 obtained by different purification methods, where ViT-B is considered**
>
> | Model           | Std ACC | Robust ACC (ℓ∞) | Robust ACC (ℓ2) |
> |------------------|---------|------------------|------------------|
> | MAE-Pure         | 84.24   | 38.27            | 49.50            |
> | MaskDiT-Pure     | 85.27   | 46.25            | 62.29            |
> | RMAE-Pure        | 84.89   | 43.33            | 58.33            |
> | **RMaskDiT-Pure**| **85.33** | **59.18**      | **70.58**        |
>
> The results show that the performance is worse compared to WRN-28. Our analysis suggests that this may be due to the fact that the ViT architecture is a data-hungry model, requiring larger datasets to fully leverage its potential.
>
> [1] Adbm: Adversarial diffusion bridge model for reliable adversarial purification.
>
> [2] Towards Understanding the Robustness of Diffusion-Based Purification: A Stochastic Perspective
>
> [3] Diffusion models for adversarial purification.

---

> > ### Comment · Reviewer_eQ1q · 2025-08-07
> >
> > Sorry for the late reply and I would like to thank the authors for their rebuttal. Most of my concerns are well-addressed and I will increase my score to 4.

---

> ### Author Response · Authors · 2025-08-04
>
> Dear Reviewer eQ1q,
>
> I hope this message finds you well. As the discussion period is nearing its end with less than three days remaining, we would like to ensure that all your concerns have been addressed satisfactorily. If there are any additional points or feedback you would like us to consider, please don’t hesitate to let us know. Your insights are invaluable to us, and we are eager to resolve any remaining issues to further improve our work.
>
> Thank you again for your time and effort in reviewing our paper.
>
> Best
>
> Regard
>
> Team 21073

---

> ### Author Response · Authors · 2025-08-06
> **Follow-up on Your Valuable Feedback**
>
> Dear Reviewer eQ1q,
>
> Thank you very much for your meaningful and constructive suggestions. We truly appreciate the time and thought you have devoted to reviewing our work.
>
> We would like to let you know that we have carefully addressed your concerns regarding the experimental setup. Specifically, we have added the missing standard deviations and clarified the runtime cost of different purification pipelines.
>
> In addition, we have included new ablation study to isolate the contribution of the proposed MaskDiT-Pure designs, and have also added a ViT-based classifier in our evaluation to enhance the generality of our findings.
>
> We hope these clarifications and additional experiments adequately address your concerns. Please don’t hesitate to let us know if you have any further comments or recommendations. We would greatly appreciate the opportunity to further discuss with you.
>
>
> Best regards,
>
> Your Friends
>
> Team 21073

---

> ### Author Response · Authors · 2025-08-07
> **With Sincere Gratitude**
>
> Dear Reviewer eQ1q,
>
> Thank you very much for your response and for taking the time to reconsider our work. We truly appreciate your thoughtful feedback and are grateful that our rebuttal was able to address your concerns. We're also sincerely thankful for your updated score and support.
>
> Best regards,
>
> Your Friends
>
> Team 21073

---

### Comment · Area_Chair_swiq · 2025-08-01

Dear Reviewers,

The author-reviewer discussion phase has started. If you want to discuss with the authors about more concerns and questions, please post your thoughts by adding official comments as soon as possible.

Thanks for your efforts and contributions to NeurIPS 2025.

Best regards,

Your Area Chair

---

### Author Response · Authors · 2025-08-09
**Summary of our rebuttal**

Dear All Reviewers,

We sincerely thank all reviewers for their thorough evaluation and professional guidance on this paper. The review comments have provided us with important directions for improvement and have significantly enhanced the completeness and rigor of our work. We will fully incorporate these suggestions in future revisions.

We would first like to provide an overall summary of our rebuttal: **For Reviewer eQ1q**, we added standard deviation and runtime cost results, along with multiple ablation studies, to verify that the performance gains stem from our core idea rather than inherent advantages of the model itself; **For Reviewer tuPn**, we provided a detailed explanation of the theoretical derivation, supplemented with new theoretical proofs, additional baseline comparisons for MAE-based methods, and a clearer articulation of our methodological motivation; **For Reviewer DP9v**, we clarified the logic behind the equivalence derivation, explained the rationale for indirectly optimizing the target via an upper bound, and supplemented our arguments with supporting literature and further explanations in the appendix, while also demonstrating the rigor of our reasoning; **For Reviewer WB7Q**, we added experimental results and analysis on the transferability of our method, discussed potential applications regarding border impact, and explored the feasibility of extending our approach to other modalities.

We would like to express our sincere gratitude again to all Reviewers, whose professional and constructive suggestions have greatly helped us improve our paper.

Hope everything goes smoothly for you as well.

Best Regards,

Team 21073

---

### Note · Authors · 2025-08-13

Dear AC and  all reviewers

We sincerely thank you for participating in the review of our paper. We greatly appreciate your valuable comments, which have provided us with significant opportunities for improvement and refinement. Below is a consolidated overview of our rebuttal addressing both experimental and theoretical feedback.


# Experimental aspect

## Robustness and Reliability Analyses (Reviewer eQ1q)

We added standard deviation and runtime analyses to verify result robustness and computational cost.

We conducted further ablation studies to ensure that performance gains stem from our AMV-based mechanism  rather than inherent advantages of the model architecture.

##  Expanded Baseline Comparisons (Reviewer tuPn)

We Enriched comparisons with MAE-based baseline, providing a comprehensive evaluation under different settings.

## Applicability and Transferability (Reviewer WB7Q)

We introduced new experiments evaluating our method’s transferability to mask CNNs, showing it performs denoising via AMV-based sensitivity, which is not transferable to non-transformer architectures.

We further analyzed its possible impact and applicability in border-related tasks.

# Theoretical aspect

## Strengthened Theoretical Foundations (Reviewer tuPn)

We incorporated the supremum of the adversarial loss, which can be used to monitor its minimal optimization.

We further discussed the potential of the MAE architecture for denoising, along with its advantages and limitations, and outlined a direction for future work.

## Equivalence and Optimization Justification (Reviewer DP9v)

We clarified the relationship between AMV and reconstruction loss, and in Theorem 3.2 highlighted that minimizing an upper bound to optimize an intractable or non-differentiable objective is a widely recognized and accepted practice.

We clarified that our approximation methods are well-established in the literature, so concerns about bound tightness or multi-approximation error analysis are not applicable.


Finally, we extend our special thanks to AC swiq for their attention to this paper and their responsible, dedicated attitude throughout the review process. We also thank all reviewers for their valuable comments, which have guided the refinement of our work; we have addressed these suggestions and will incorporate the improvements into the revised version.

Hope everything goes smoothly for you as well.

Best Regards,

All authors of Team 21073

---

### Decision · Program_Chairs · 2025-09-17

**Decision:**

Reject

**Comment:**

This paper offers a new perspective on adversarial purification by focusing on preserving inter-patch semantic consistency through attention matrix analysis—a direction largely unexplored in prior work. Initial reviews mainly focus on further evaluation of the proposed method by adding more experiments and the motivation to use MAE. The authors addressed the former one during the rebuttal but failed to address the latter one. After the discussion, reviewers are still concerned about the choice of MAE and are not sure of the key motivation and insights behind this choice, which is the key reason for this borderline case (no reviewer support for the acceptance, eventually all borderline acceptance). Given the remaining concerns of this paper, I recommend that the authors prepare another version to better justify the selection of MAE and discuss the potential to extend the proposed method to a framework level.